# Differentiation of primate primordial germ cell-like cells following transplantation into the adult gonadal niche

Enrique Sosa[1], Di Chen[1], Ernesto J. Rojas[1], Jon D. Hennebold[2,3], Karen A. Peters[4], Zhuang Wu[5], Truong N. Lam[5], Jennifer M. Mitchell[6], Meena Sukhwani[4], Ramesh C. Tailor[7], Marvin L. Meistrich[5], Kyle E. Orwig[4], Gunapala Shetty[5] & Amander T. Clark [1]

A major challenge in stem cell differentiation is the availability of bioassays to prove cell types generated in vitro are equivalent to cells in vivo. In the mouse, differentiation of primordial germ cell-like cells (PGCLCs) from pluripotent cells was validated by transplantation, leading to the generation of spermatogenesis and to the birth of offspring. Here we report the use of xenotransplantation (monkey to mouse) and homologous transplantation (monkey to monkey) to validate our in vitro protocol for differentiating male rhesus (r) macaque PGCLCs (rPGCLCs) from induced pluripotent stem cells (riPSCs). Specifically, transplantation of aggregates containing rPGCLCs into mouse and nonhuman primate testicles overcomes a major bottleneck in rPGCLC differentiation. These findings suggest that immature rPGCLCs once transplanted into an adult gonadal niche commit to differentiate towards late rPGCs that initiate epigenetic reprogramming but do not complete the conversion into ENO2-positive spermatogonia.

[1] Department of Molecular, Cell and Developmental Biology, Eli and Edythe Broad Center of Regenerative Medicine and Stem Cell Research, University of California, Los Angeles, Los Angeles, CA 90095, USA. [2] Division of Reproductive and Developmental Sciences, Oregon National Primate Research Center, Beaverton, OR 97006, USA. [3] Department of Obstetrics and Gynecology, Oregon Health & Science University, Portland, OR 97239, USA. [4] Department of Obstetrics, Gynecology and Reproductive Sciences and Magee Women's Research Institute, University of Pittsburgh School of Medicine, Pittsburgh, PA 15213, USA. [5] Department of Experimental Radiation Oncology, University of Texas MD Anderson Cancer Center, Houston, TX 77030, USA. [6] Department of Veterinary Medicine and Surgery, University of Texas MD Anderson Cancer Center, Houston, TX 77030, USA. [7] Department of Radiation Physics, University of Texas MD Anderson Cancer Center, Houston, TX 77030, USA. Correspondence and requests for materials should be addressed to A.T.C. (email: clarka@ucla.edu)

Germline cells are essential for fertility and passing DNA from one generation to the next. In each generation, germ cell development begins around the time of embryo implantation with the differentiation of founding progenitors called primordial germ cells (PGCs). PGCs are transient and in the appropriate environment will subsequently advance in differentiation towards oogonia in females and pro-spermatogonia in males. In an inappropriate environment, however, the latent pluripotency program can be reactivated leading to germ cell tumors including teratomas. Moreover, the abnormal specification of PGCs has the potential to impact the quality of the entire cohort of germ cells in the adult gonad given that after PGC specification no other cell type can contribute to the germline. Therefore, understanding the biology of PGCs has important implications for future reproductive success and child health.

One of the most exciting models for understanding human PGC development is the pluripotent stem cell model and differentiation into PGC-like cells (PGCLCs) in vitro[1–5]. Directed differentiation protocols for generating human PGCLCs (hPGCLCs) result in the formation of so-called early PGCs which are equivalent to PGCs at around week 3 of human embryo development. Early PGCs in the primate cynomolgus (cyno) macaque are triple positive for SOX17, PRDM1, and TFAP2C, while being negative for the late stage PGC markers VASA and DAZL[6]. A recent study has demonstrated that female human embryonic stem cell (hESCs) can differentiate into VASA-positive human oocyte-like cells[7]. However, an approach for differentiating male primate PGCLCs into more advanced VASA positive stages is lacking.

Advanced differentiation and generation of fertilization competent sperm from mouse PGCLCs (mPGCLCs) was first shown by transplantation of mouse aggregates and mPGCLCs into the testicles of infertile male mice[8–10]. Furthermore, mPGCLCs have been differentiated entirely in vitro using co-culture with gonadal somatic cells[11]. The differentiation of male mPGCLCs entirely in vitro depended first upon the success of testicular transplantation to prove mPGCLC competency. In humans, transplanting hPGCLCs into the testicles of human subjects as a first-line experiment to prove hPGCLC competency is inconceivable. Instead, we propose that a first approach could instead utilize the testicular xenotransplantation bioassay or alternatively homologous transplantation of nonhuman primate PGCLCs.

Testicular xenotransplantation involves transplantation of primate (human or nonhuman) testicular cells containing germ cells into the seminiferous tubules of busulfan-treated or irradiated immune-deficient nude mice[12–16]. More recently, it was also shown that rhesus macaque PGCs (rPGCs) and human PGCs (hPGCs) can also persist and form colonies at the basement membrane of this model, indicating that the testicular xenotransplantation approach can be extended to characterize less mature germline cells, and possibly PGCLCs[17]. In all reported cases of xenotransplantation, human and nonhuman primate germ cells do not differentiate into haploid sperm in the mouse seminiferous tubule niche. Instead, they recapitulate many of the characteristics that are unique to male germline stem cells. These include the ability to (1) migrate to the basement membrane of seminiferous tubules, (2) divide to produce chains of cells with spermatogonial characteristics (a high nuclear to cytoplasmic ratio and intercellular bridges), and (3) persist for long periods of time.

In order to confirm that the testicular xenotransplantation bioassay could be used as an important reporter for germline competency despite the lack of apparent differentiation, Hermann and colleagues[18,19] showed that homologous transplantation of rhesus macaque testicular cells into recipients depleted of spermatogonial stem cells prior to transplantation promotes

spermatogenesis from donor cells[16,20]. Furthermore, not only were the donor SSCs competent to undergo complete spermatogenesis the donor-derived sperm were competent to fertilize rhesus macaque oocytes and give rise to donor-derived embryos[20]. It is unknown how less mature rhesus macaque germ cell types will respond in this assay.

In the current study, we differentiated riPSCs to early (VASA negative) rPGCLCs that we characterize as being similar to bona fide embryonic rPGCs younger than 28 days of embryo development post-fertilization. Following xenotransplantation into irradiated nude mice or homologous transplantation into irradiated rhesus macaques, we show that the seminiferous tubules environment supports the survival of rPGCLCs and promotes further differentiation to a VASA-positive state. Taken together, transplantation to the seminiferous tubule environment promotes rPGCLC differentiation beyond what can currently be achieved in vitro.

## Results

**Rhesus PGCLCs equate to rPGCs younger than embryonic day 28.** To date, all directed differentiation strategies to generate primate PGCLCs create VASA negative cells equivalent to early-stage PGCs[1,2]. In the cynomolgus macaque, VASA protein expression is induced in cynoPGCs at around Carnegie Stage (CS) 12 (Days 26–30)[6]. To evaluate this in rhesus macaque embryos, we collected time-mated CS12 rhesus embryos ($n = 3$) (Fig. 1a). Examination of transverse histological sections in the region of the aorta–gonad–mesonephros (Fig. 1a, box), revealed gonadal ridge epithelium, (Fig. 1b, arrow-heads), but no definitive gonad. Immunofluorescence (IF) staining for the transcription factor SOX17, a classical visceral endoderm marker that is also required for hPGC specification[1] revealed that SOX17 positive cells can be found in the dorsal aorta (DA), the mesonephros (m), the genital ridge epithelium (white arrow-heads), and the dorsal mesentery (white arrow) (Fig. 1c). IF staining of the transcription factor TFAP2C, a second marker of primate PGCs, revealed that TFAP2C positive cells are restricted only to the dorsal mesentery and the genital ridge epithelium (Fig. 1d, box and arrow), and not the DA. Therefore, the DA SOX17 positive cells are most likely the hemogenic endothelium[21]. This also indicates that TFAP2C is a discriminating marker of primate PGCs in this region of the embryo at this developmental stage[6]. Triple IF staining of SOX17, TFAP2C, and the transcription factor PRDM1, all markers of early PGCs, identifies triple positive rPGCs in the dorsal mesentery, as well as rPGCs closely associated with the gonadal ridge epithelium (Fig. 1d). Furthermore, all TFAP2C positive rPGCs in CS12 co-express the pluripotent transcription factor OCT4 (Fig. 1e). To determine whether rPGCs at CS12 correspond to early rPGCs (VASA negative) or late rPGCs (VASA positive)[6], we stained for VASA and the rPGC marker TFAP2C. Our data indicate that the majority of CS12 rPGCs were located in either the dorsal mesentery or the genital ridge epithelium, and the majority of TFAP2C positive rPGCs were also VASA positive (Supplementary Figure 1a, b). In addition, rare VASA negative TFAP2C positive cells were also identified (Fig. 1e, white arrows). As expected SOX2 was not expressed in any TFAP2C positive rPGCs (Fig. 1e).

Once we determined that the antibodies raised against TFAP2C, SOX17, PRDM1, OCT4, and VASA (Supplementary Table 1) could discriminate rPGCs in the embryo, we next examined the expression of these proteins with differentiation of riPSCs into rPGCLCs. The rPGCLC differentiation strategy involved a modification[5] of the two-step differentiation protocol first described by Sasaki and colleagues[2]. The first step involves harvesting undifferentiated riPSCs cultured on mouse embryonic

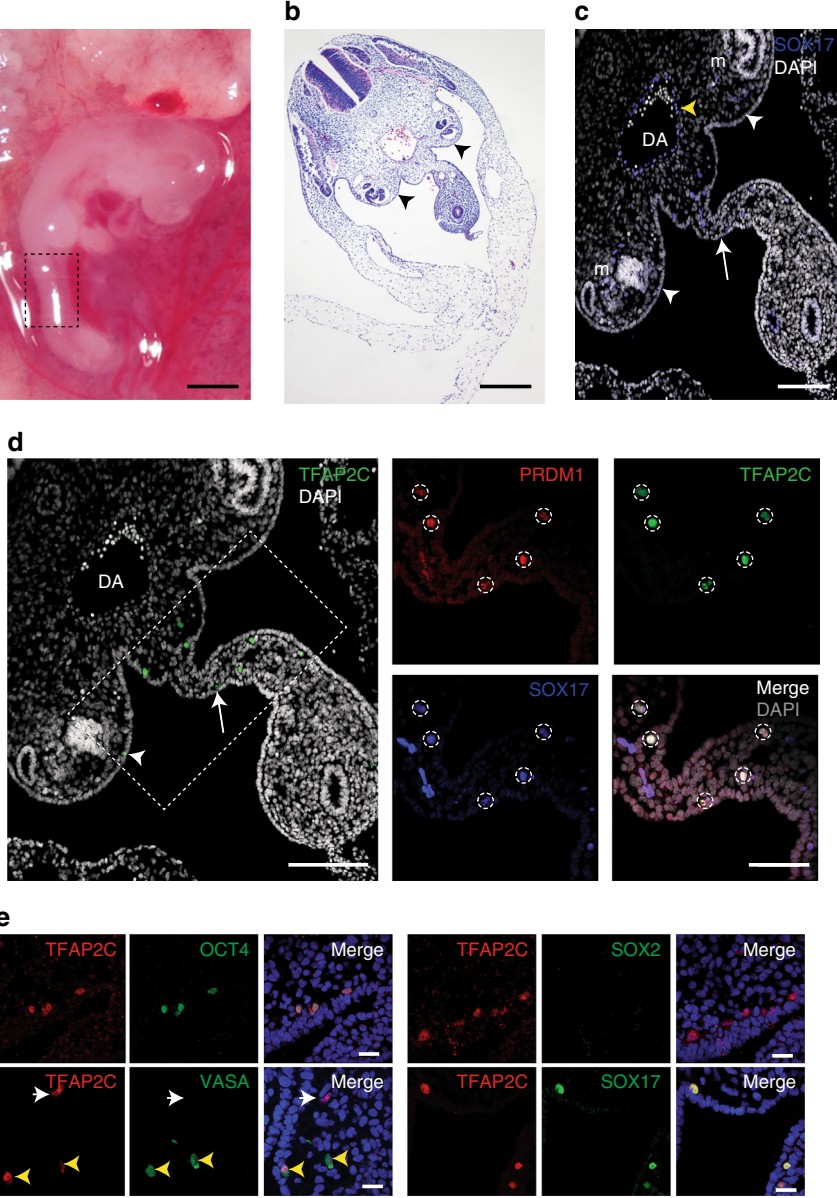

**Fig. 1** Identification of rPGCs by immunofluorescence using early and late rPGC markers. **a** CS12 rhesus macaque embryo. Box highlights the region that was assessed for migrating rhesus Primordial Germ Cells (rPGCs) Scale bar, 1 mm. **b** Transverse histological section of CS12 embryo reveals the genital ridge epithelium (arrowheads) Scale bar, 400 μm. **c** Immunofluorescence (IF) staining of CS12 sections showing SRY-box 17 (SOX17, blue) and DAPI nuclear stain (white) in genital ridge epithelium (white arrowheads), adjacent mesonephros (m) dorsal mesentery (white arrow), or hemogenic endothelium (yellow arrowhead) within the dorsal aorta (DA). Scale bar, 200 μm. **d** IF staining of AP-2 gamma (TFAP2C, green) can discriminate rPGCs (box) in the dorsal mesentery (arrow) as well as the genital ridge epithelium (arrowhead) of a CS12 embryo. Triple IF staining of PR/SET domain 1 (PRDM1, red), SOX17 (blue), and TFAP2C (green) can be used to mark rPGCs (dotted circles) (merge, white; DAPI nuclear stain, gray). Scale bar, 200 μm. **e** Dual IF for TFAP2C (red) confirms co-expression with the germ cell marker OCT4, SOX17, and VASA. Yellow arrow-heads indicate TFAP2C/VASA double positive rPGCs. White arrowheads indicate TFAP2C positive VASA negative rPGCs. SOX2 (green) is not expressed in rPGCs. Merged images show DAPI nuclear stain (blue). Scale bars, 15 μm. $n = 3$ CS12 embryos, shown are images from Day (D) 28

fibroblasts (MEFs) as a single cell suspension followed by differentiating the riPSCs for 24 h to create incipient mesoderm-like cells (iMeLCs) (Fig. 2, Supplementary Figure 2a). The second step involves differentiating the iMeLCs as three-dimensional aggregates in low adhesion 96-well plates (Fig. 2a). Rhesus PGCLCs are formed in step 2 within the aggregates in response to bone morphogenetic protein 4 (BMP4). In the first experiment, we used IF to test whether the transcription factors SOX17, PRDM1, or TFAP2C are expressed in undifferentiated riPSCs, or iMeLCs prior to aggregate differentiation. We also examined the expression of BRACHYURY (BRA) to confirm

iMeLC induction, as well as pluripotency transcription factors OCT4 and SOX2. We found that both riPSCs and riMeLCs expressed OCT4 and TFAP2C, and in addition, riMeLCs also expressed BRA as expected[2]. SOX2 was uniformly expressed in riPSCs and heterogeneously in iMeLCs. Importantly, we found that riPSCs, and iMeLCs did not expressed SOX17 or PRDM1 (Supplementary Figure 2b, c). Therefore, we hypothesize these transcription factors could be used together to document the emergence of nascent rPGCLCs in the aggregate.

To identify rPGCLCs, aggregates were assessed at Day (D) 1 to D4 for SOX17 and PRDM1 expression. SOX17/PRDM1 double

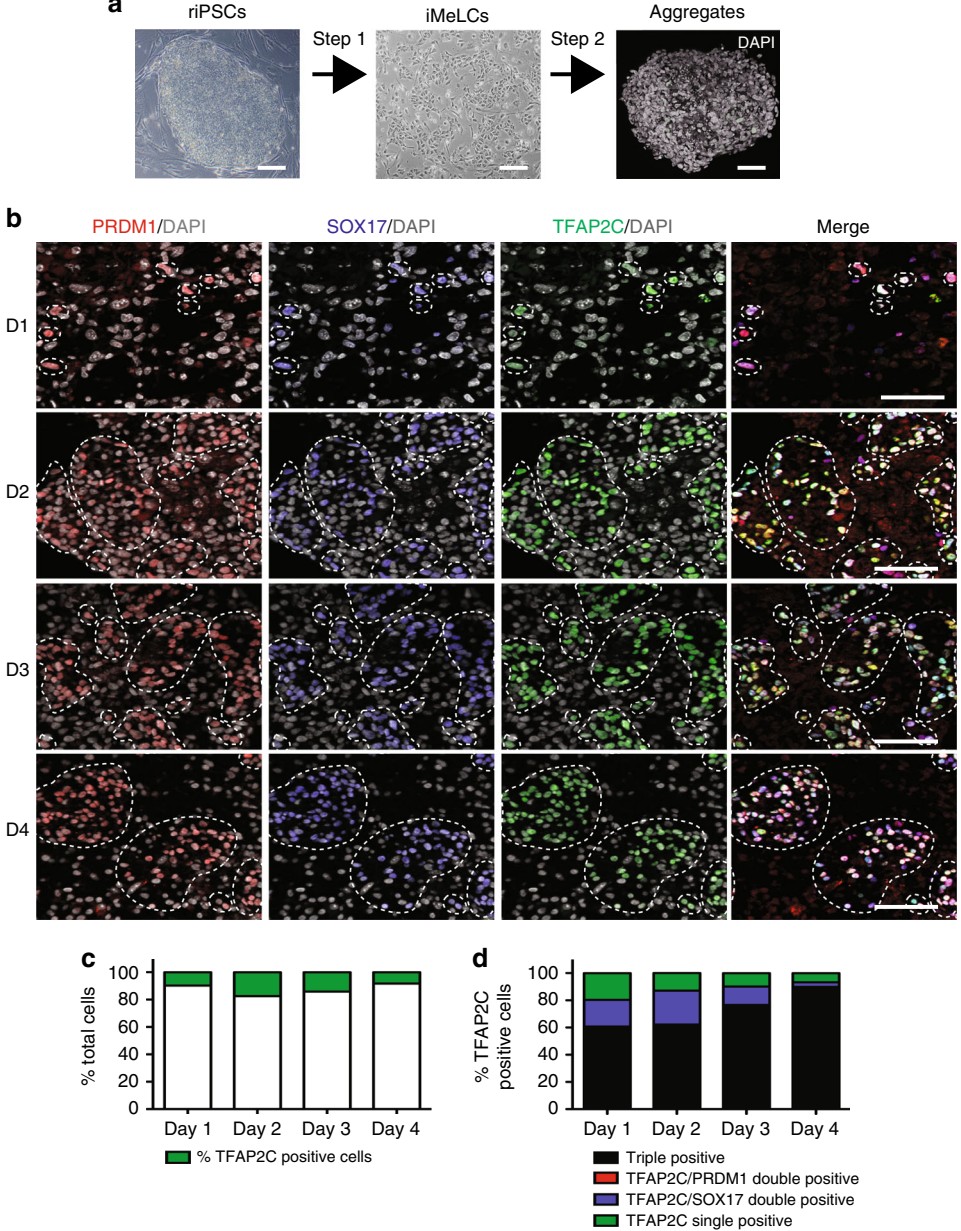

**Fig. 2** Specification of rPGCLCs from riPSCs. **a** Schematic overview of rPGCLC-specification in aggregates via an iMeLC intermediate. Scale bars = 100 μm, shown are images of riPSC89 differentiation. **b** Triple IF staining for germ cell markers during riPSC89 aggregate differentiation from Day (D) D1 through D4. PRDM1 (red), SOX17 (blue), TFAP2C (green), DAPI (gray). Dotted lines highlight TFAP2C positive cells (single and clusters). Scale bars, 50 μm. $N = 3$ biological replicates. **c** The percent of TFAP2C positive cells (green) in aggregates at D1–D4. **d** The percent of TFAP2C positive cells in aggregates at D1–D4 that are either single (TFAP2C, green), double (TFAP2C/PRDM1, red or TFAP2C/SOX17, blue), or triple (TFAP2C/PRDM1/SOX17, black) positive for early rPGC markers. Shown are results using the riPSC90 cell line, unless otherwise stated

positive cells were identified starting on D1 through D4 with only a rare single positive cells (Supplementary Figure 3a). Previous studies analyzing cyno embryos at the time of lineage specification identified SOX17/PRDM1 double positive cells as marking both cynoPGCs as well as visceral endoderm (VE) cells[6]. Therefore, to discriminate between these two possibilities in the aggregates, we also stained SOX17 together with TFAP2C, which is positive in cynoPGCs while being negative in VE[6]. Double IF staining of aggregates revealed that subpopulations of SOX17 positive cells were also TFAP2C positive at D1, with more double positive cells emerging from D2 (Supplementary Figure 3b).

Based upon the SOX17/TFAP2C dual staining results, we examined rPGCLC formation by using the triple stain to identify

cells co-expressing PRDM1/SOX17/TFAP2C in the aggregates. Using this approach, we discovered triple positive rPGCLCs as early as D1 of aggregate differentiation in two independent lines of riPSCs (riPSC89 and riPS90) (Fig. 2b and Supplementary Figure 3c). These triple positive rPGCLCs persisted through D4 and up to D8 (Fig. 2b and Supplementary Figure 3c). However, by D15 of aggregate differentiation, triple positive rPGCLCs were undetectable in either riPSC line (Supplementary Figure 3c–e).

Given that TFAP2C may be expressed earlier in the rhesus macaque relative to hPGCs, we quantified the total percent of TFAP2C positive cells from D1 to D4 of aggregate differentiation, and discovered that TFAP2C positive cells are a minor fraction of the total aggregate at each time point (Fig. 2c). Furthermore,

TFAP2C positive cells were mostly triple positive at all time points (TFAP2C together with SOX17 and PRDM1). However TFAP2C/SOX17 double positive cells were clearly identified notably at D1–3 whereas TFAP2C/PRDM1 double positive cells were never identified (Fig. 2d). Taken together, our results indicate that triple positive PRDM1/SOX17/TFAP2C rPGCLCs are induced at D1 of aggregate differentiation, and that the potential rPGCLC precursors are either SOX17/TFAP2C or SOX17/PRDM1 double positive but not TFAP2C/PRDM1.

**Rhesus macaque PGCLCs arrest before epigenetic reprogramming**. To identify the transcriptional identity of rPGCLCs, we isolated rPGCLCs using fluorescence activated cell sorting (FACS) at D1, 2, 4, and 8 of aggregate differentiation using antibodies that recognize EPCAM and ITGA6 (Fig. 3a). EPCAM and ITGA6 were chosen because they were previously used to recognize hPGCLCs in aggregate differentiation[2]. The frequency of EPCAM/ITGA6 double positive cells was highest in D2 aggregates, while at D4, this population stabilized at around 9.5%. On D8, the percent of EPCAM/ITGA6 ultimately constitutes only a small fraction of all cells in an aggregate (Fig. 3a). Using real-time polymerase chain reaction (PCR) to detect rPGC markers NANOS3, TFAP2C, PRDM1, and cKIT, we show that the rPGCLC genes are all up-regulated in the EPCAM/ITGA6 double positive population compared to undifferentiated riPSCs and riMeLCs (Fig. 3b).

To determine whether rPGCLCs are in the early or late stages of rPGC development, we examined VASA expression using IF at D4 and D8 and discovered that all TFAP2C positive rPGCLCs in aggregates are VASA negative (Fig. 3c). In contrast, late rPGCs at CS12 (D28) and CS23 (D50) are VASA positive (Fig. 3c).

Prior to the transitioning from early to late, cynoPGCs undergo global loss of 5-methyl cytosine (5mC)[6]; to examine this, we performed IF using antibodies that recognize 5mC together with OCT4 to identify rPGCLCs/rPGCs. OCT4 positive rPGCs at CS12 (D28) and CS23 (D50) are negative for 5mC as expected (white circles). In contrast, rPGCLCs at D2, D4, and D8 of aggregate differentiation are all positive for 5mC (Fig. 3d). This result suggests that in aggregates, the rPGCLCs have not undergone global DNA methylation reprogramming as detected using IF. In addition to 5mC, we also evaluated 5-hydroxymethylcytosine (5hmC), which is dynamically enriched in the mPGC genome during mPGC migration and genital ridge colonization[23–26]. In the rhesus embryos at CS12 (D28), we discovered that rPGCs were positive for 5hmC, albeit at low levels and similar to somatic cells of the embryo. In contrast at CS23 (D50), the rPGCs and most somatic cells were 5hmC negative. In the aggregates, all rPGCLCs were 5hmC positive (Fig. 3e). Taken together, rPGCLCs generated in vitro correspond to SOX17/TFAP2C/PRDM1 newly specified rPGCs that have not initiated global 5mC/5hmC epigenetic reprogramming as monitored by IF.

**Xeno- and homologous transplantation promote differentiation**. Previously, it was reported that CS23 rPGCs xenotransplanted into the seminiferous tubules of the busulfan-treated mouse testicles colonize the seminiferous tubule basement membrane giving rise to colony-like chains of cells[17]. To determine whether rPGCLCs persist in this assay and form colonies, we first created a green fluorescent protein (GFP) expressing subline of riPSC89 using a lentivirus (referred to as riPSC89$^{UbiC:GFP}$). Prior to transplantation, karyotype analysis was performed to confirm a normal 42XY male karyotype[27]. The riPSC89$^{UbiC: GFP}$ line was differentiated through the two-step protocol ending at D8 of aggregate differentiation. The D8 aggregates were shipped overnight on cold packs to MD Anderson Cancer Center in Houston

where they were dissociated and xenotransplanted into the seminiferous tubules of adult immune-deficient nude mice that had been irradiated to ablate endogenous spermatogenesis. The recipient mouse testes were seeded with between $5.0 \times 10^4$ and $1.3 \times 10^5$, unsorted aggregate cells per transplant ($n = 6$ testicles). As a control, we also transplanted $1.9 \times 10^5 - 2.7 \times 10^5$ undifferentiated riPSCs per transplant ($n = 6$ testicles). Eight weeks after transplant the mice were euthanized, and the testicles were analyzed.

At the time of dissection, 6/6 recipient testicles receiving a single cell suspension of aggregate cells were GFP positive. In contrast, only 4/6 recipient testicles that received undifferentiated riPSCs were GFP positive (Supplementary Figure 4a). This result is consistent with previous reports that single cell suspensions of undifferentiated primate pluripotent stem cells exhibit poor survival, which our data would suggest extends to the seminiferous tubule environment. Whole mount IF of the GFP positive testicles using the non-human primate (Nhp), anti-rhesus macaque testis cell antibodies[18] followed by AlexaFluor488 secondary antibodies demonstrated that the Nhp/GFP positive cells in the recipient testicles that received aggregates were VASA positive indicative of rPGCLC differentiation. In contrast, the recipient testicles that received undifferentiated riPSCs were VASA negative (Fig. 4a). Histological analysis of the GFP positive recipient testicles xenotransplanted with donor aggregates sometimes revealed cysts (4/6) (Supplementary Figure 4b–e, asterisk) but, did not form teratomas. In contrast, testicles transplanted with riPSCs yielded teratomas in 2 out of 4 of cases.

To confirm that xenotransplantation supported rPGCLC differentiation, sections of testicles were processed to paraffin and sections stained for VASA together with the spermatogonial stem cell markers MAGEA4 and ENO2. Using these markers, we show that almost all Nhp positive rPGCLCs in the recipient mouse testicular tubules differentiate into VASA positive germline cells confirming the results of the whole mount. Furthermore, these Nhp-positive germline cells were exclusively located on the basement membrane, typical of spermatogonia. A fraction (28%) of NHP positive rPGCLCs also expressed the spermatogonial protein MAGEA4 (Fig. 4b). In contrast, ENO2 was not expressed in any Nhp-positive cells ($n = 15$) in the mouse testis (Fig. 4b).

Due to the finding that only one of the spermatogonial stem cell markers (MAGEA4) was expressed in xenotransplanted donor rPGCLCs, we hypothesized that the rPGCLCs may not correspond to adult-stage spermatogonia and instead the MAGEA4 single positive germ cells may represent an earlier stage of primate germ cell development. To address this, we stained human fetal testis at 17 weeks post-fertilization and found ENO2/MAGEA4 double positive hPGCs (hPGCs were identified with VASA) (Supplementary Figure 4f). In contrast, at earlier embryonic points (D28 and D50), MAGEA4 was expressed in VASA positive rPGCs whereas ENO2 was not (Supplementary Figure 4g). Given that MAGEA4 and ENO2 were not expressed in rPGCLCs before xenotransplantation (Supplementary Figure 4g), our results support the hypothesis that xenotransplantation results in rPGCLC differentiation into VASA/MAGEA4 positive rPGCLCs corresponding to a stage in embryo development prior to ENO2 positive spermatogonial differentiation.

To confirm that the rPGCLCs originated from the specified rPGCLCs in the aggregates and not de novo specification of rPGCLCs donor aggregate cells while in the seminiferous tubules, we used FACS to isolate EPCAM/ITGA6 positive (rPGCLCs) and negative (somatic) cells, and xenotransplanted the two populations separately into the testicles of irradiated SCID mice (Supplementary Figure 4h). Using confocal microscopy of intact testicles, we identified GFP-positive signal only in the testicles xenotransplanted

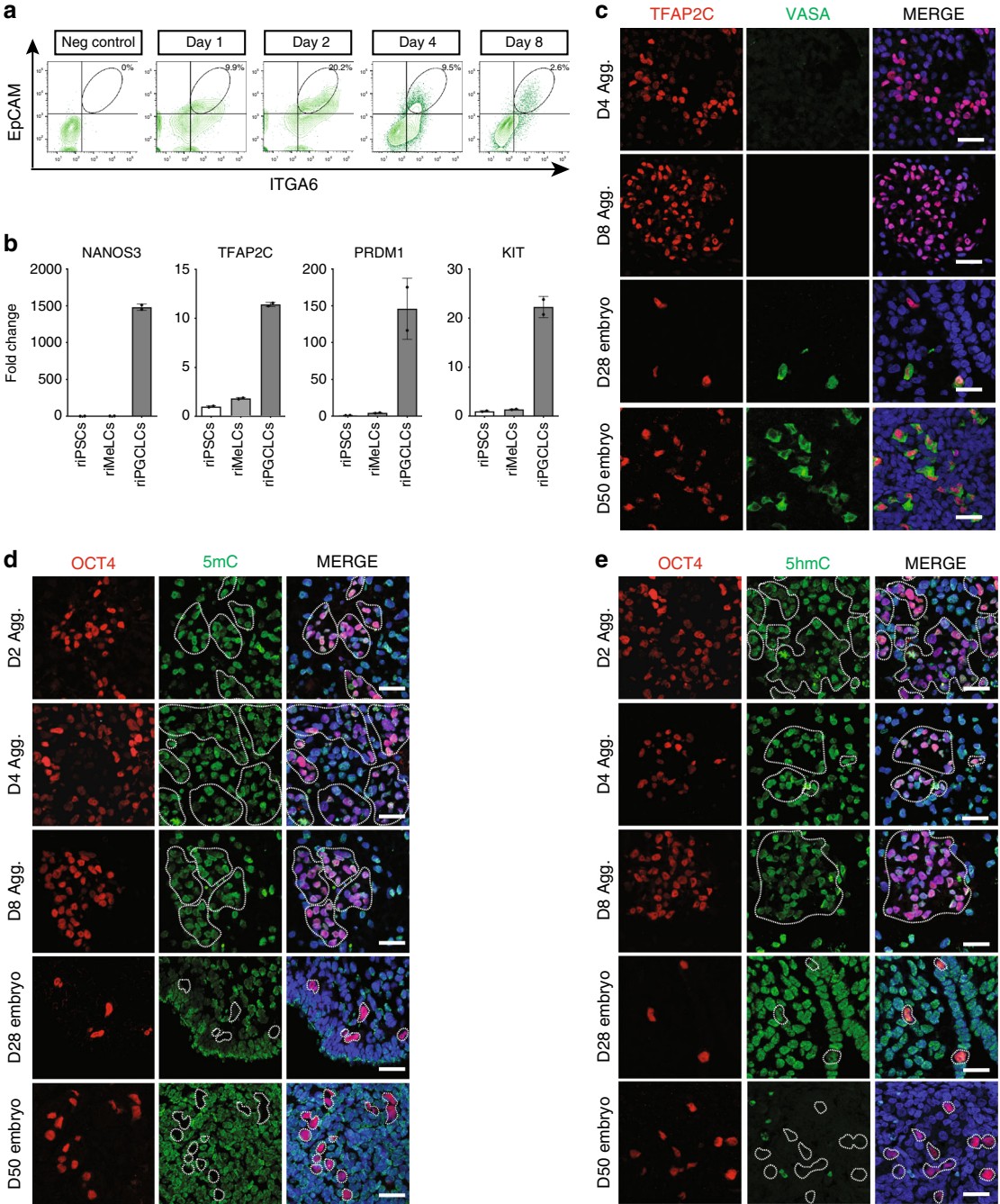

**Fig. 3** rPGCLCs in aggregates correspond to early rPGCs prior to global 5mC erasure. **a** FACS plots showing the percent of EPCAM/ITGA6 double positive cells in undifferentiated riPSC89 (neg control), and riPSC89 aggregates at Day (D) 1, 2, 4, and 8. **b** RT-PCR analysis show enrichment of early rPGC markers in rPGCLCs (riPSC89) isolated by FACS at D6 using EPCAM/ITGA6. All data points used to generate the bar graphs (black circles) were overlaid to the corresponding bar graphs, error bars represent the Standard Error of the Mean (S.E.M.). **c** IF staining of rPGCLCs/rPGCs identified with TFAP2C (red) indicate that the late-stage rPGC marker VASA (green) is not expressed in rPGCLCs at D4 or D8 and instead is expressed in rPGCs in CS12 (D28) and CS23 (D50). Scale bars, 20 μm. **d** IF staining showing 5mC (green) is present in OCT4-positive(+) (red) rPGCLCs in aggregates (dotted white line) at D2, D4, and D8, while being absent from OCT4+ (red) rPGCs (dotted white line) in CS12 (D28) and CS23 (D50) embryos. Scale bars, 20 μm. **e** IF staining of 5hmC (green) in OCT4+ (red) rPGCs (dotted white line) at CS12 (D28) as well as in OCT4+ (red) rPGCLCs (dotted white line) in D2, D4, and D8 aggregates. CS23 (D50) rPGCs (OCT4+, red) do not have detectable 5hmC (green). Scale bars, 20 μm. Shown are results using riPSC90 unless otherwise stated. $N = 3$ biological replicates for riPSCs and CS12 (D28); $n = 2$ biological replicates for CS23 (D50)

with FACS isolated rPGCLCs but not in testicles xenotransplanted with somatic cells or sham control (no donor cells) (Supplementary Figure 4h). This result suggests that the rPGCLCs in the xenotransplant differentiated from the rPGCLCs originally specified in the aggregates, which subsequently underwent further differentiation following xenotransplantation.

Based on the induction of MAGEA4 and VASA in the xenotransplanted donor cells, the expectation is that the colonized rPGCLCs should also have initiated epigenetic reprogramming. To address this, we transplanted a single cell suspension of D4 aggregates from the riPSC90 line into busulfan-treated nude mice and examined enrichment of 5mC and 5hmC using IF. Given that

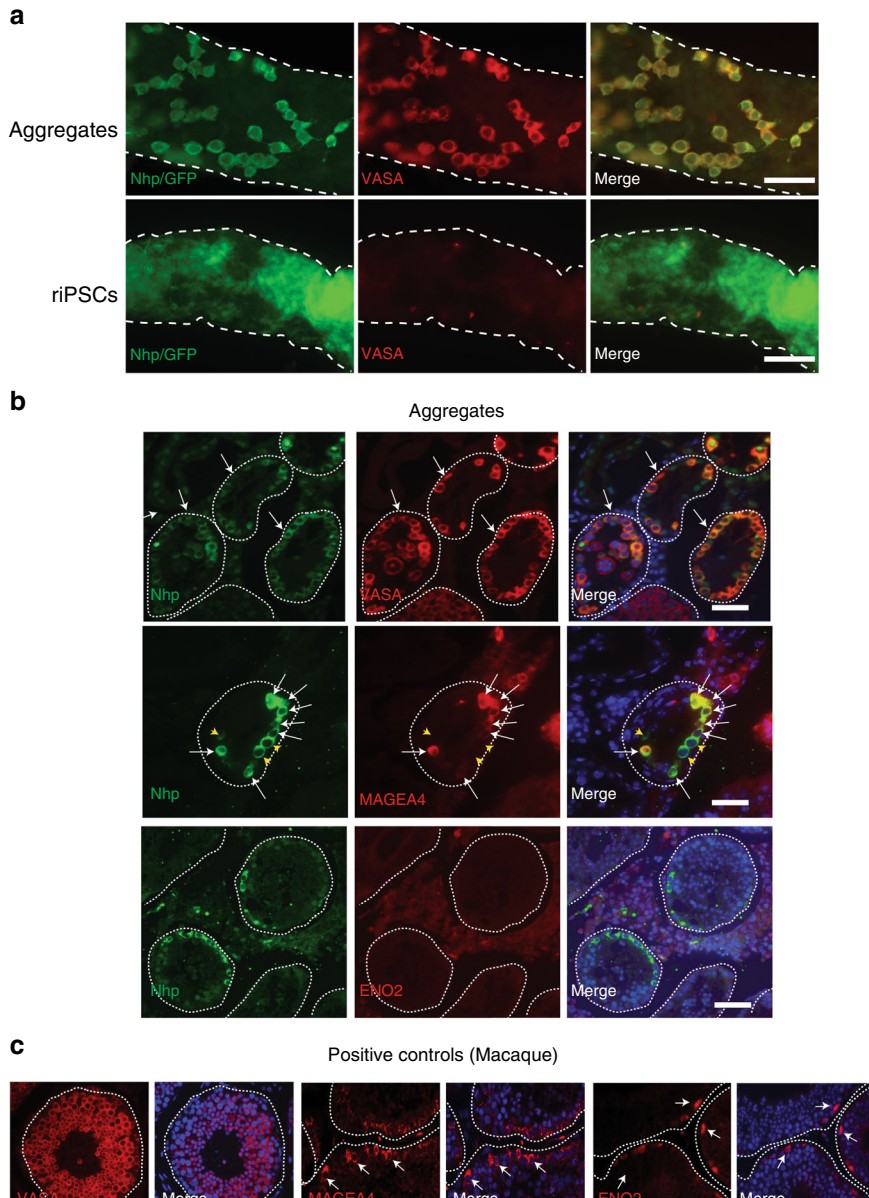

**Fig. 4** Xenotransplantation of rPGCLCs leads to expression of late stage markers. **a** Whole mount immunofluorescence (IF) staining of mouse testicle seminiferous tubules xenotransplanted with a single cell suspension of Day (D) D8 riPS89<sup>UbiC:GFP</sup> aggregates, reveals the presence of Nhp/GFP (green) cells, near the basement membrane of the tubules (white dotted outline) that co-express the late-stage rPGC marker VASA (red). Recipient testicles xenotransplanted with riPSCs contained Nhp/GFP positive cells but did not express VASA. Scale bars, 40 μm. **b** IF staining on paraffin-embedded, testicle sections from recipients that received a single cell suspension of aggregate cells. Nhp-positive cells (green) are found inside the seminiferous tubules (basement membrane, white dotted outline) and express VASA (white arrows). A subset of Nhp-positive cells expressed MAGEA4 (red, white arrows), while ENO2 was absent. Scale bars, 40 μm. **c** Positive control staining for VASA (red), MAGEA4 (red), and ENO2 (red) in adult rhesus macaque testis. Scale bars, 40 μm

riPSC90 was not labeled with GFP, the donor rPGCLCs were identified using an antibody that detects the primate-specific, nuclear mitotic apparatus protein (NuMA). This result shows that the NuMA positive rPGCLCs undergo global loss of 5mC (Fig. 5a). To detect 5hmC in these xenotransplants, we performed sequential IF staining of adjacent sections, and show that NuMA positive rPGCLCs are also 5hmC positive (Supplementary Figure 5a–d). Therefore, this result supports the hypothesis that xenotransplantation induces rPGCLC differentiation to the equivalent of late-stage of rPGCs in the middle of epigenetic reprogramming similar to those identified at CS12, but younger than those at CS23 where 5hmC and 5mC are no longer detectable by IF.

Given that the loss of 5hmC in mPGCs occurs largely through replication-coupled demethylation, we were also interested in determining whether rPGCs are in cycle during this developmental window. To achieve this we stained for the nuclear antigen Ki67 in rhesus embryos at CS12 (D28) and CS23 (D50) as well as aggregates at D4 and D8 (Fig. 5b). In all cases, the rPGCLCs/rPGCs were identified using IF for TFAP2C (red) with Ki67 labeled in green. The results are quantified as the percentage of Ki67 positive cells in the TFAP2C positive or negative population (Fig. 5c). During aggregate differentiation, the TFAP2C positive rPGCLCs at D4 are mostly Ki67 negative (not in cycle), whereas the somatic cells are Ki67 positive. In contrast

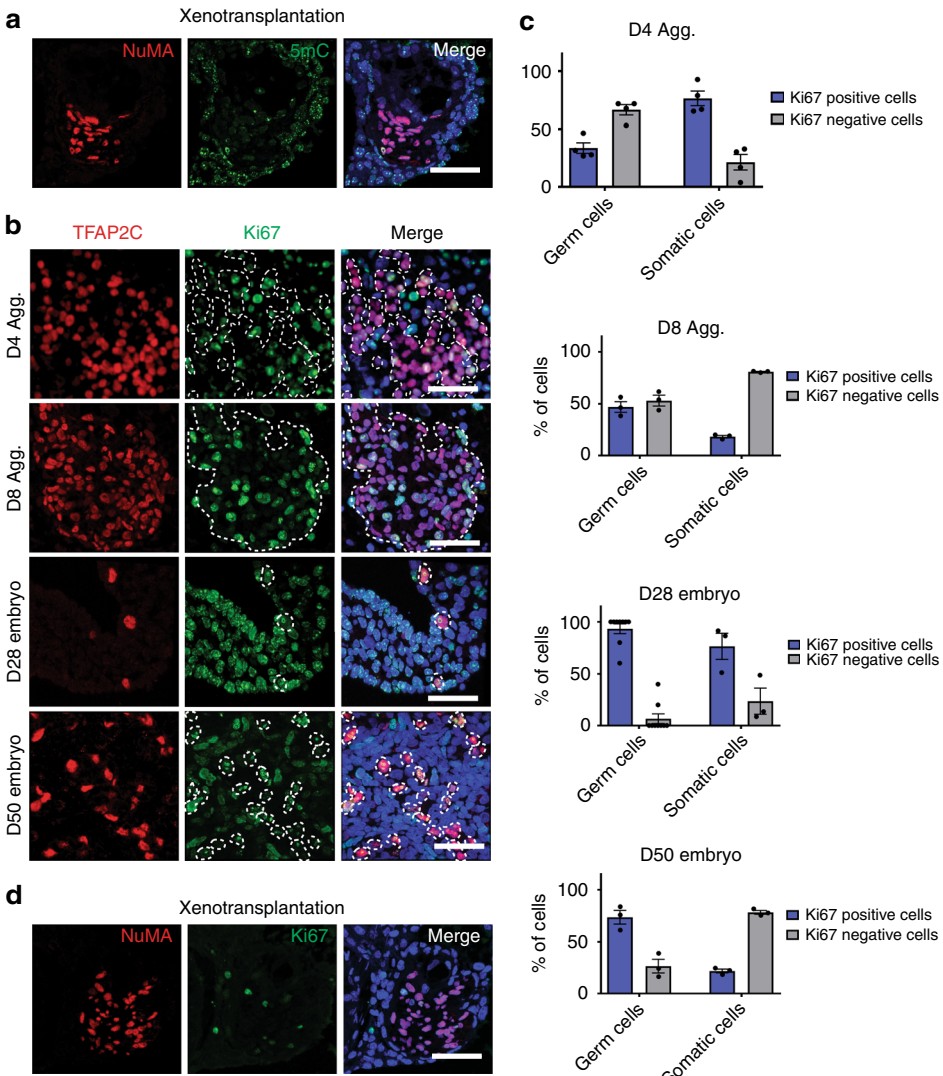

**Fig. 5** Xenotransplantation is associated with rPGCLCs epigenetic reprograming and precocious exit from the cell cycle. **a** Donor cells are detected through their expression of the nuclear mitotic apparatus protein (NuMA, red) in paraffin sections of recipient testicles ($n = 13$) xenotransplanted with a single cell suspension of Day (D) D4 aggregate cells derived from riPSC90. 5mC (green) was detected in recipient somatic cells. Scale bar, 50 μm. **b** IF on paraffin sections of aggregates at D4 and D8, and rhesus embryos at CS12 (D28) and CS23 (D50) for the rPGCLC/rPGC marker TFAP2C (red) and the proliferation marker, Ki67(green). Nuclei were detected using DAPI (blue). Scale bars, 50 μm. **c** Quantification of Ki67 positive (blue bar graphs) and Ki67 negative (gray bar graphs) rPGCLCs/rPGCs and somatic cells in aggregates at D4 and D8, and rhesus embryos at CS12 (D28) and CS23 (D50). All data points used to generate bar graphs (black circles) were overlaid to the corresponding bar graphs, error bars represent the Standard Error of the Mean (S.E. M.), $N = 3$ technical replicates. **d** Recipient mouse testicles ($n = 13$) xenotransplanted with riPSC90 D4 aggregate cells co-stained for NuMA (red) and Ki67 (green). Nuclei were detected using DAPI (blue). Scale bar, 50 μm

at D8, the majority of somatic cells are Ki67 negative, whereas 50% of the rPGCLCs are now in cycle and are Ki67 positive. In the embryo, rPGCs at D28 and D50 are almost Ki67 positive and therefore in the cell cycle during 5mC/5hmC epigenetic reprograming. Analysis of the xenotransplants shows that the majority of NuMA positive donor cells (red) are Ki67 negative and therefore the rPGCLCs are not in cycle following xenotransplantation (Fig. 5d). Taken together, our data indicate that xenotransplantation induces rPGCLC differentiation and epigenetic remodeling, however the adult testicular niche causes the rPGCLCs to exit the cell cycle before completing epigenetic reprograming and global removal of 5hmC.

Due to the inability of primate cells to fully reprogram and differentiate to ENO2 positive spermatogonia in the mouse testicular niche, we next tested the hypothesis that homologous transplantation of aggregates into irradiated rhesus macaque male

testicles could remove the barrier to rPGCLC differentiation. Therefore, we transplanted dissociated aggregates containing rPGCLCs or undifferentiated riPSCs into the left and right testis, respectively, of rhesus macaque males depleted of germ cells by irradiation ($n = 2$). The monkeys were also administered GnRH-antagonist treatment starting immediately after irradiation for 2 months until the time of transplantation. GnRH-antagonist treatment is expected to facilitate survival and colonization of transplanted cells[16]. Serum testosterone levels were monitored to confirm transient suppression of testosterone during GnRH antagonist treatment and maintenance of Leydig cell function after irradiation (Fig. 6a). Ultrasound-guided rete testis injection was performed under general anesthesia to transplant 4.2 million single aggregate cells, and 9.8 million undifferentiated riPSCs into the rete testis. Following transplantation, testicular volumes and serum testosterone were measured monthly in each animal

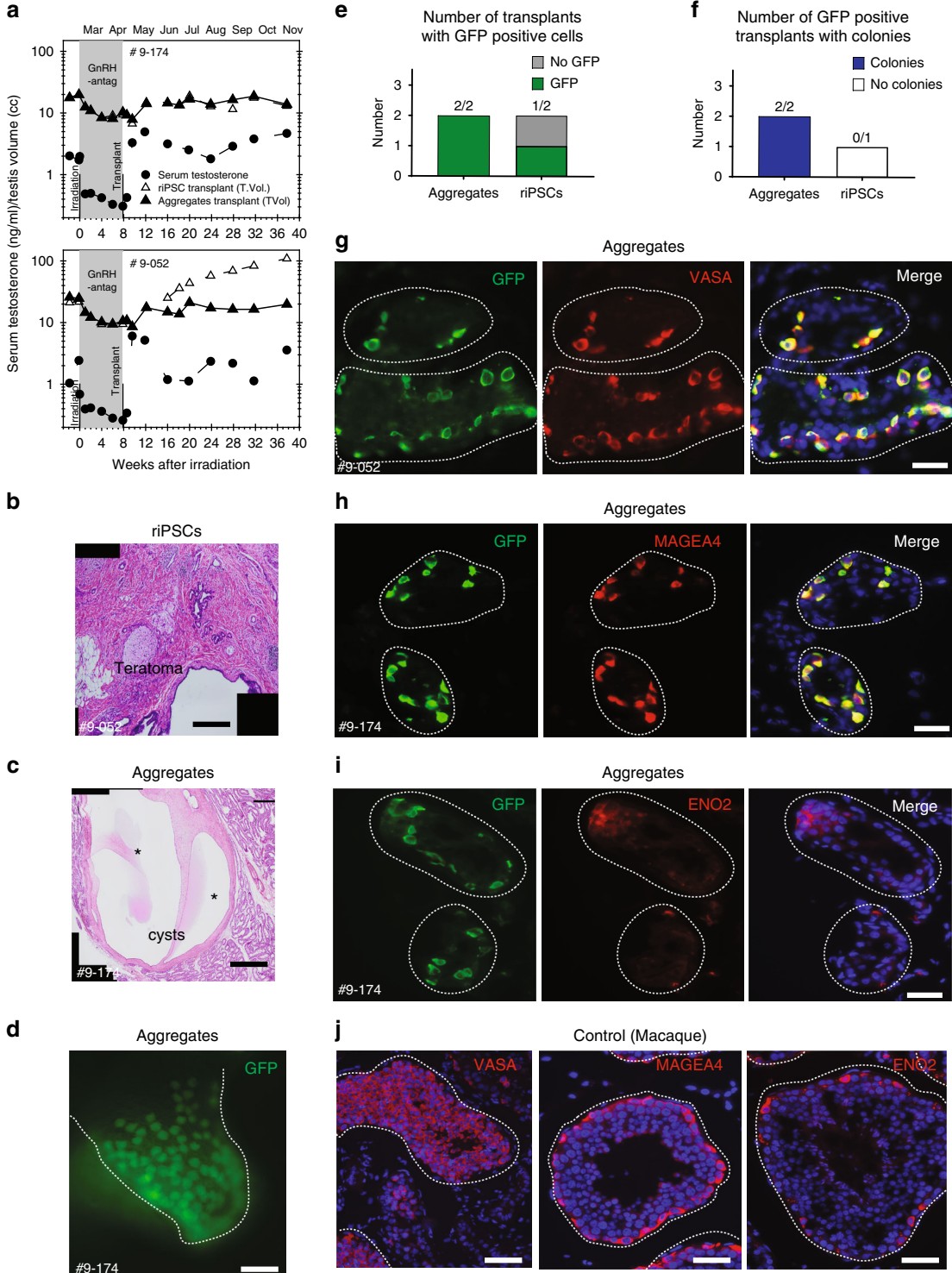

**Fig. 6** Homologous transplant of rPGCLCs results in differentiation to late rPGC stage. **a** The volume of recipient testicles that received donor Day (D) D8 riPSC89[UbiC:GFP] aggregate cells returned to baseline levels, comparable to the volume prior to irradiation and subsequent GnRH antagonist treatment. In contrast one recipient male who received riPSC89[UbiC:GFP] cells exhibited a volume increased. **b** H&E staining of the abnormally large recipient testicle transplanted with riPSC89[UbiC:GFP] revealed teratomas. Scale bars, 400 μm. **c** H&E staining of aggregate recipient testicles reveals large cysts (asterisks) near normal tubules. Scale bars, 400 μm. **d** Expression of GFP in recipient testicles transplanted with D8 riPSC89[UbiC:GFP] aggregate cells within the seminiferous tubules (white dotted lines). Scale bars, 40 μm. **e** A graph of the number of transplants that had GFP positive donor cells. **f** Graph of the number of GFP positive transplants with colonies. **g–j** IF on sections of rhesus recipient testicles that received a single cell suspension of aggregate donor cells (GFP positive, green). The surviving donor cells co-expressed the late stage rPGC markers **g** VASA (red) and **h** MAGEA4 (red), but not **i** ENO2. Nuclear staining using DAPI (blue) is shown in merged panels for **g–i**. Scale bars, 40 μm. **j** Positive IF staining of rhesus adult testicle for VASA (red), MAGEA4 (red), or ENO2 (red) in combination with the nuclear marker DAPI (blue). Scale bars, 40 μm

(Fig. 6a). In animal 9-174, the final volume and weight of recipient testicles at 7 months after transplantation were indistinguishable from each other, despite one testicle receiving aggregate cells, whereas the other testicle received undifferentiated riPSCs. In contrast, in the other animal (animal number 9-052), the testicle transplanted with undifferentiated riPSCs was 11-fold heavier relative to the contra-lateral testicle transplanted with aggregate cells (Fig. 6a).

Given the abnormal growth of the testicle in animal 9-052 that received undifferentiated iPSCs, the experiment was terminated at 7 months after transplant, and the testicles from both animals were removed. Upon dissecting the testicles from animal 9-052, it was apparent under visual inspection that the testicle receiving aggregate cells contained cysts, whereas the testicle that received undifferentiated riPSCs was filled with solid tumorigenic masses (Supplementary Figure 6a, 6b). It was also possible to identify GFP positive signal in both testicles from 9-052 (Supplementary Figure 6c). Histological analysis confirmed that transplantation of undifferentiated riPSCs resulted in the formation of a teratoma in animal 9-052, which had almost completely invaded the testicle (Fig. 6b).

In order to determine whether rPGCLCs generated MAGEA4/ENO2 positive spermatogonial stem cells, the testicles from 9-174 and 9-052 were fixed, embedded, and serially sectioned. Using GFP to mark the transplanted cells, we identified rare GFP positive cells along the seminiferous tubule basement membrane of rhesus macaque testicles injected with aggregates (2/2 transplants of aggregates) (Fig. 6d, e) that resulted in the formation of colonies (Fig. 6f). In contrast, the GFP-positive testicle transplanted with undifferentiated riPSCs exhibited no colonies (Fig. 6f). The GFP positive cells did not represent complete spermatogenesis, and instead were organized as individual cells, and occasionally pairs. Moreover, we show that the GFP positive cells along the basement membrane were positive for VASA and MAGEA4 but were negative for ENO2 (Fig. 6g–i). Control rhesus macaque seminiferous tubules confirmed antibody specificity for VASA, MAGEA4, and ENO2 (Fig. 6j). Taken together, these findings confirm homologous transplantation of aggregates containing rPGCLCs leads to differentiation into VASA positive, MAGEA4 positive cells, however the adult testicular niche does not support differentiation into ENO2 positive spermatogonia.

## Discussion

Here, we establish the rhesus macaque as another non-human primate species amendable to rPGCLC differentiation in vitro. Consistent with previous studies in the human[1,2] and in the cynomolgus macaque[4], VASA RNA and protein are not detected in rPGCLCs during aggregate formation[28]. VASA is an evolutionary conserved gene that is expressed in germline cells of all metazoan, and in mammals is expressed as PGCs approach the developing genital ridge[6,29,30]. In the current study, we show that a genital ridge is not necessary to induce VASA protein expression in rPGCLCs, and instead VASA protein can be induced following xeno or homologous transplant into an adult testicular niche.

The mechanism responsible for inducing VASA protein in mammalian PGCs is unclear. Using transgenic mouse technology, ectopic expression of mouse vasa homologue (mvh) in somatic cells can be achieved by the global loss of DNA methylation[31]. In the current study, our data show that differentiation into VASA-positive rPGCLCs is accompanied by a global loss of 5mC. Therefore, we hypothesize that the expression of VASA in rPGCLCs is associated with DNA methylation reprogramming following xenotransplantation. Given that rPGCLCs are lost between D8 and D15 of aggregate differentiation in vitro, we

propose that it is the transfer of rPGCLCs to a new environment supportive of rPGCLC survival that is primarily responsible for triggering global DNA demethylation and advancing differentiation into VASA-positive cells. Consistent with this, the culture of mPGCs/mPGCLCs in growth factors and chemicals that maintain mPGC survival also enable the transition to MVH-positive state without a requirement for testicular cells[32,33]. Therefore, it is likely that adult testicular cells per se are not responsible for inducing VASA expression. Given this result, we anticipate that chemically defined conditions could be identified in the future to support the differentiation of rPGCLCs without the requirement for transplantation.

The markers MAGEA4 and ENO2 were originally chosen as an approach to examine rPGCLC differentiation to adult spermatogonial stem cells following xenotransplantation. Given the lack of ENO2 expression in MAGEA4 positive transplant-derived rPGCLCs, we hypothesized that MAGEA4 may be marking an earlier stage of germ cell differentiation when ENO2 is negative. Here, we show that MAGEA4 is expressed much earlier in rhesus germ cell development, being present in VASA positive rPGCs at CS12 as they approach the genital ridge epithelium, as well as in VASA positive rPGCs at CS23. In contrast ENO2 is not expressed in rPGCs at these stages, instead being expressed together with MAGEA4 and VASA in putative pro-spermatogonia. It is difficult to predict whether the lack of ENO2 in the transplant-derived rPGCLCs is due to a block in developmental timing, or whether ENO2 is abnormally silenced. However, failure to also erase 5hmC in the xenotransplant-derived rPGCLCs, together with exit from the cell cycle is consistent with a block in developmental timing such that the transplanted rPGCLCs are arresting in development and exiting the cell cycle at a stage equivalent to rPGCs younger than CS23.

Previous studies showed that testicular xenotransplantation of undifferentiated hESCs and hiPSCs into the testicles of busulfan-treated nude mice causes differentiation into VASA positive hPGCLCs accompanied by teratoma, embryonal carcinoma, and yolk sac tumor formation[34]. The tumors in previously published studies most likely originate from hiPSCs/hESCs because testicular xenotransplantation of embryonic testicles containing bona fide hPGCs results in colony formation without tumorigenesis[17]. Unlike studies using human pluripotent stem cells, we did not identify any evidence of VASA positive putative rPGCLCs in the testicles transplanted with undifferentiated riPSCs. The underlying reason for this discrepancy is unclear and may be due to species-specific differences in the donor cells (human verse rhesus).

Successful resumption of spermatogenesis from donor cells in homologous monkey-to-monkey transplants involves the transfer of tissues containing adult spermatogonia[18,20]. In the current study, the lack of spermatogenesis in homologous transplants could be due to either an incompetent rPGCLC or alternatively the possibility that rPGCLC/rPGCs are incapable of differentiating in an adult niche. In support of the young verses old niche hypothesis, mouse transplantation studies using neonatal infertile recipients can support spermatogenesis from mPGCs, whereas adult infertile recipients can support spermatogenesis from pro-spermatogonia or spermatogonia[9]. Given this, the only way to experimentally test whether rPGCs have the competency to fully differentiate into haploid sperm in an adult niche would be to perform homologous transplants of bona fide rPGCs into infertile rhesus recipients at different ages (from neonate to adult). The feasibility of this theoretical experiment is challenged by the very low numbers of rPGCs found during the embryonic stages of interest.

In summary, we show that rPGCLCs differentiated in aggregates are competent to differentiate into VASA/MAGEA4

positive rPGCLCs capable of epigenetic reprogramming and global loss of 5mC when transplanted into an adult testicular niche. However, our data also indicate that transplantation into the adult niche is associated with rPGCLCs exiting from the cell cycle before the loss of 5hmC can be completed. Given an equivalent stage of rPGCLC differentiation was achieved in homologous and xenotransplants, our studies also suggest that testicular xenotransplantation may be a cost-effective option for testing lineage commitment of in vitro specified primate PGCLCs prior to conducting costlier, and time-consuming non-human primate, autologous or homologous transplantation experiments. Although it is well known that full reconstitution of spermatogenesis cannot be achieved in the mouse from non-human primate cells, our study highlights the similarities when either mice or monkeys are used as recipients of in vitro differentiated rPGCLCs.

## Methods

**Time-mated breeding of rhesus macaque**. Time-mated breeding of rhesus macaque males and females was performed by measuring estradiol daily in the female starting from D5 to D8 after menses began[17]. Pregnancy was confirmed by measuring progesterone as well as by ultrasound. Embryos at CS12 were collected by C-section ($n = 3$ at D28, $n = 1$ at D26, $n = 1$ at D30). All rhesus macaque time-mated breeding experiments were conducted following the approval of the Oregon National Primate Research Center (ONPRC) Institutional Animal Care and Use Committee (IACUC).

**Maintenance and directed differentiation of riPSC lines**. Undifferentiated riPSC89[27] and riPSC90[22] cells were cultured on a feeder layer of mitomycin C-treated MEFs in hESC media (DMEM/F-12) (Life Technologies), 20% KSR (Life Technologies), 10 ng/mL bFGF (R&D Systems), 1% nonessential amino acids (Life Technologies), 2 mM L-glutamine (Life Technologies), Primocin™ (Invivogen), and 0.1 mM β-mercaptoethanol (Sigma). Media was changed daily and colonies were passaged manually every 5 days. For iMeLC inductions we followed previously published protocols with minor modifications[2,5]. Briefly, D5 riPSCs colonies were trypsinized (0.05% trypsin) (Life Technologies), resuspended in MEF media. The MEFs were depleted by plating the cell suspension in tissue culture dishes, two times, for 5 min each. The resulting cell suspension was pelleted and resuspended in iMELC media (GMEM) (Life Technologies), 15% KSR (Life Technologies), 0.1 mM nonessential amino acids (Life Technologies), penicillin/streptomycin/L-glutamine (Life Technologies), Primocin™ (Invivogen), 0.1 mM β-mercaptoethanol (Sigma), sodium pyruvate (Life Technologies), activin A (PeproTech), CHIR99021 (Stemgent), Y-27632 (Stemgent), filtered through a 40 μm cell strainer (Falcon) and plated at a density of $1.0 \times 10^5$ cells per well of a human plasma fibronectin (Invitrogen)-coated 12-well plate. After 24 h of incubation at 37 °C with 5.0% $CO_2$, iMeLCs were trypsinized (0.05% trypsin, Life Technologies) and resuspended in PGCLC media (GMEM) (Life Technologies), 15% KSR (Life Technologies), 0.1 mM nonessential amino acids (Life Technologies), penicillin/streptomycin/L-glutamine (Life Technologies), Primocin™ (Invivogen), 0.1 mM β-mercaptoethanol (Sigma), sodium pyruvate (Life Technologies), 10 ng/mL human LIF (EMD Millipore), 200 ng/mL BMP4 (R&D Systems), 50 ng/mL EGF (Fisher Scientific), 10 μM Y-27632 (Stemgent), and plated at a density of $3.0 \times 10^3$ cells per well of a low adherence spheroid forming 96-well plate (Corning). Aggregates were collected for analysis on Days 1, 2, 3, 4, 8, and 15 of directed PGCLC differentiation.

**Human fetal samples**. Human prenatal testes were acquired following elected termination and pathological evaluation after UCLA-IRB review which deemed the project exempt under 45 CRF 46.102(f). All prenatal testes were obtained from the University of Washington Birth Defects Research Laboratory (BDRL), under the regulatory oversight of the University of Washington IRB approved Human Subjects protocol combined with a Certificate of Confidentiality from the Federal Government. BDRL collected the fetal testes and shipped them overnight in HBSS with an ice pack for immediate processing in Los Angeles. All consented material was donated anonymously and carried no personal identifiers. Developmental age was documented by BDRL as days post-fertilization using a combination of prenatal intakes, foot length, Streeter's Stages, and crown-rump length. All prenatal gonads documented with a birth defect or chromosomal abnormality were excluded from this study.

**Immunofluorescence (IF) staining**. Aggregates containing rPGCLCs were collected and then fixed in 4% PFA and embedded in histogel (Thermo Scientific) to facilitate subsequent embedding into paraffin blocks. Sections of aggregates, rhesus embryos, or human fetal tissues (5 μm) placed onto microscope slides were then de-paraffinized and rehydrated through a xylene, ethanol series. For antigen retrieval, slides were heated to 95 °C in Tris-EDTA solution (10 mM Tris Base,

1 mM EDTA solution, 0.05% Tween-20, pH 9.0). Sections were permeabilized (PBS, 0.05% Triton-100) and then blocked in PBS containing 10% normal donkey serum. The primary antibodies anti-5mC (AMM99021; 1:100), anti-5hmC (39769; 1:100), anti-5hmC (51660S; 1:100), anti-OCT4 (sc8628x; 1:100), anti-OCT3/4 (sc5279; 1:100), anti-TFAP2C (sc12762; 1:200), anti-TFAP2C (sc8977; 1:200), anti-PRDM1 (9115S; 1:100), anti-SOX17 (GT15094; 1:100), anti-SOX2 (MAB2018; 1:100), anti-VASA (AF2030; 1:100), anti-Brachyury/T (AF2085; 1:200), anti-MAGEA4 (Clone 57B; 1:30), anti-ENO2 (MMS-518P; 1:500), anti-Ki67 (556003; 1:200), anti-NuMA (Ab84680; 1:200) (Supplementary Table 1) were incubated overnight at 4 °C. This was then followed by incubating the slides for 60 min at room temperature with their corresponding, species-specific, secondary antibodies: donkey anti-mouse IgG AF488 (715-546-150; 1:500), donkey anti-rabbit IgG AF488 (A21206; 1:500), donkey anti-goat IgG AF488 (705-546-147; 1:200), goat anti-mouse IgG$_{2a}$ AF488 (A21131; 1:200), donkey anti-goat IgG AF594 (705-586-147; 1:200), donkey anti-mouse IgG$_{2b}$ AF594 (A21145; 1:200), donkey anti-rabbit IgG AF594 (711-585-147; 1:500), donkey anti-mouse IgG AF594 (A21203; 1:200), donkey anti-goat IgG AF647(A21447; 1:200), donkey anti-rabbit IgG AF647 (A31573; 1:500) (Supplementary Table 1). Mounting media (Prolong-gold anti-fade w/DAPI, Invitrogen) was added and slides were sealed. Slides were allowed to cure for at least 24 h at 4 °C prior to imaging.

riPSCs and riMeLCs were plated onto culture slides (Corning) and grown overnight in their respective media, at 37 °C with 5.0% $CO_2$. Cells were washed with PBS then fixed in 4% PFA for 10 min. Cells were rinsed with PBST and then permeabilized (PBS, 0.05% Triton-100) for 10 min. Nonspecific binding was blocked (PBS, 10% normal donkey serum for 30 min) and then incubated overnight at 4 °C with primary antibodies (Supplementary Table 1). Species-specific secondary antibodies were added to slides and incubated at room temperature for 1 h. Mounting media was added, slides were sealed, and then cured for 24 h.

Mouse testes tissues were fixed overnight in 4% paraformaldehyde, 5 μm thick paraffin sections were used for the immunofluorescence study. The monkey testes were fixed in cold 4% paraformaldehyde overnight, processed through a sucrose gradient, and then embedded in OCT compound; 8 μm cryosections were cut for IF staining. For testicular histology of both mouse and monkey, Bouin's or 4% paraformaldehyde-fixed tissues were embedded in paraffin and sections were stained in periodic acid-Shiff's-hematoxylin.

For fluorescent immunostaining, either the deparaffinized or frozen sections were subjected to antigen retrieval by initially heating in boiling citrate buffer (BioGenex) for 2 min followed by cooling for 30–60 min. The slides were rinsed and blocked in antibody diluent. Subsequently, sections were stained with the following primary antibodies in antibody diluent: goat anti-VASA (AF2030; 1:100), mouse anti-ENO2 (MMS-518P; 1:500), mouse anti-MAGE-A4 (Clone 57B, kindly provided by Dr. Giulio Spagnoli, University Hospital Basel, Switzerland; 1:30), rabbit anti-GFP (2956S; 1:100), rabbit anti-rhesus testis cell (NHP) (provided by Kyle Orwig; 1:200). For double immunofluorescent staining, the two primary antibodies incubated with the tissue sections were detected with species-specific secondary antibodies (Supplementary Table 1). Stained sections were mounted with VectaShield mounting media containing DAPI (Vector Laboratories) and imaged. Positive immunoreactivity was validated by the omission of primary antibody and testing the antibody in a tissue in which it is known to be positive.

**Microscopy**. Confocal images of the riPSC89 line, riPSC90 line, iMeLCs, and sectioned aggregates were examined on an LSM 880 (Carl Zeiss) with a Plan-Apochromat 20×/0.8 NA and a Plan-Apochromat 63×/1.4 NA M27 oil immersion objective at room temperature. Acquired images were processed using IMARIS 8.1 (Bitplane). H&E slides were examined on an Olympus BX-61 light microscope. Images were processed with Image J version 1.51d (NIH). D28 embryos, teratomas, and cysts images were stitched together using the grid-stitching plugin in Image J version 1.51d (NIH).

For immunofluorescence imaging of tissue sections or whole mounts of seminiferous tubules, a Leica DM 4000B (Leica Microsystems) microscope was used.

**Image analysis**. Quantification of confocal images was performed using IMARIS 8.1 (Bitplane) microscopy image analysis software. To quantify the total percent of cells that were positive for TFAP2C in D1, D2, D3, and D4 aggregates, we first quantified the total number of nuclei using the spot detection function to detect DAPI. Only DAPI positive nuclei that overlapped with TFAP2C signals were counted, using the IMARIS spot function. Using the TFAP2C channel and the co-localization function in IMARIS we built co-localization channels for SOX17 and PRDM1 channels. Through this strategy, we were able to quantify single, double, and triple positive cells found in aggregates using the IMARIS spot function. Three random sections were counted for each sample for Days 1–4 of differentiation and these experiments were repeated on two separate cell lines (riPSC89 and riPSC90) with at least 2 technical replicates each. Graphs were made using Prism 7 (Graph Pad) data analysis software.

To quantify the total percent of cells that were positive (+) for Ki67, first we counted the total number of nuclei using the spot detection function to detect DAPI. Next, utilizing TFAP2C to demark germ cells we built a co-localization channel for Ki67. The spot function was used on the newly built co-localization

channel to quantify the number of cells that were both positive for TFAP2C and Ki67. To quantify the number of somatic cells that were Ki67$^+$, we counted the total number of cells that were positive for Ki67 and subtracted from the total number of Ki67$^+$ germ cells (TFAP2C$^+$). Three random fields were counted for each sample with 3 replicates each for Days 1–4 of differentiation. Graphs were made on Prism 7 (Graph Pad) and error bars represent the standard error of the mean.

**Fluorescence activated cell sorting analysis of rPGCLCs**. The rPGCLCs at D1, D2, and D4 were dissociated using 0.05% trypsin, while D8 PGCLCs were dissociated using 0.05% trypsin and Collagenase IV. Cells were washed with MEF media and then re-suspended in FACS buffer. Dissociated cells were incubated with anti-ITGA6-BV421 (BioLegend 313624; 1:50) and EPCAM-PE (Life Technologies A15782; 1:50) antibodies. Double positive cells were collected using an ARIA-H Fluorescence Activated Cell Sorter. Cells were sorted into RLT buffer and stored at −20 °C until ready to isolate RNA. Sorts of rPGCLCs were performed on at least 2 replicates for each group. Cytometry analysis was performed using FlowJo™ version 10.

**Establishment of the riPSC89$^{UbiC:GFP}$ reporter line**. The riPSC89 GFP-reporter line was established through episomal delivery of the lentivirus GFP-IRES_PUR-O_cassette, a kind gift from Dr. Zoran, using lipofectamine 3000 (Invitrogen). Karyotypes performed by Cell Line Genetics (Madison, WI) indicated that the GFP-expressing riPSC89 line had a normal karyotype.

**Whole mount immunofluorescence of seminiferous tubules**. For quantitative analysis of donor rhesus testis cell colonization, intact seminiferous tubules were prepared from nude mouse recipient testes, collected 8 weeks after transplantation[16]. Donor-derived colonies of spermatogonia were detected in intact seminiferous tubules by whole mount immunofluorescent staining with the rhesus testis cell antiserum[18]. Samples were dehydrated stepwise in methanol and then incubated in MeOH:DMSO:H$_2$O$_2$ (4:1:1) for 2 h. The rhesus testis-cell antibody was used and detected with rabbit secondary antibodies (Supplementary Table 1). Samples were mounted with Vectashield medium containing DAPI (Vector Laboratories) on slides with raised coverslips and visualized by fluorescence microscopy.

In co-staining experiments, the rhesus testis cell and VASA antibodies were detected with species-specific antibodies (Supplementary Table 1).

**RT-PCR**. Purified RNA was extracted from undifferentiated riPSCs, riMeLCs, and D6 rPGCLC sorted cells (EPCAM/ITAG6) using the RNAeasy mini kit (Qiagen). RNA was converted to cDNA using Superscript II Reverse Transcriptase (Invitrogen). Gene expression analysis was performed using the following gene expression assays: *TFAP2C*: Rh02844868_m1 (Applied Biosystems); *PRDM1*: Rh02837836_m1 (Applied Biosystems); *KIT*: qMccCIP0023334 (BioRad); *NANOS3*: qMccCEP0031201 (BioRad); *GAPDH*: qMccCIP0038690 (BioRad). Context sequences for the probes used were provided by the manufacturer and are as follows:

TFAP2C:
GCGATTGTTTTGGGGGACGCCGGACGCCATGTTGTGGAAAATAACTGA
TAATGTCAAGTACGAAGAGGACTGCGAGGATCGCCACGACGGGAGCAG
CAATGGAAATCCGCGAGTCCCCCACCTCTCCTCCGCCGGGCAGCACCTC
TACAGCCCCGCGCC

PRDM1:
CGCCAAGGTGCGCGTCTGTACGGCTCAGCCCGGCGGGGGACGCGGGGA
GAATGTGGACTGGGTAGAGATGAGCGAGACTTTTCTCAGATGTTGGA
TATTTGCTTGGAAAAACGTGTGGGTACGACCTTGGCTGCCCCCAAGTG
TAGCTCCAGCACTGTGAGGTTTCAGGGATTGGCAGAGGGGACCAAGGG
GACCATGAAAATGGACATGGAGGATGCGGATATGACTCTGTGGACAGA
GGCTGAGTTTGAAGAGAA

KIT:
TAGTCAACGTTGCCTGACGTTCATAATTGAAGTCACCGTGATGCCAGC
TATTATATTTCTCCTGTAGTTTAGTCTGACTGTTTTCTCTTTTCCACGTT
GAGTACACAGAACTAGACACATCTTTTATTGTGCATGTCACTG

NANOS3:
TCATCGGTCCCAGTGCCAGGATCCAAGGATCAGAAGCGCAGCCTGGAG
TCCTCGCCAGCTCCCGAACGCCTGTGCTCTTTCTGCAAACACAACGGCG
AGTCCCGGGCCATCTACCAGTCCCACGTGCTGAAGGACGAGGC

GAPDH:
GAGACACCATGGGGAAGGTGAAGGTCGGAGTCAACGGATTTGGTCGTA
TTGGGCGCCTGGTCACCAGGGCTGCTTTTAACTCTGGTAAAGTGGATA
TTGTTGCCATCAATGACCCCTTCATTGACCTCAACTACATGGTTTACAT
GTTCCAGTATGATTCCACCCATGGCAAGTTCCATGGCACCGTGAAGGCT
GAGAACGGGAAGCT

**Xenotransplantation into the seminiferous tubules**. A single cell suspension of D8 aggregates (0.1 mg/mL Collagenase, Type IV (Gibco)) and then 0.05% trypsin (Gibco) and undifferentiated riPSCs (0.1 mg/mL Collagenase, Type IV (Gibco)) were created prior to xenotransplantation. The single cell suspension was injected

in the right and left testicles of nude mice depleted of germ cells by irradiation using a $^{137}$Cs gamma-ray unit. The radiation was delivered as an initial 1.5-Gy dose and followed by a second dose of 12 Gy[35,36]. About 7 μL of solution (a range of $5.4 \times 10^4$–$2.7 \times 10^5$ cells) was injected per testicle[12]. Mice were allowed to recover and sacrificed 2 months later. In most cases, testicles were fixed in 4% paraformaldehyde, paraffin-embedded and sectioned for IF analysis. In some cases, seminiferous tubules were dispersed using collagenase and DNase, fixed in 4% paraformaldehyde and prepared for the whole mount IF staining. For injection of riPS90 into busulfan-treated nude mice, a 7 μL solution of aggregate cells ($7.0 \times 10^5$–$8.0 \times 10^4$ cells) was injected into the rete testes according to previously published approaches[17]. For transplantation of FACS isolated riPSC89$^{UbiC:GFP}$ rPGCLCs, mice were irradiated as above, and $1.1 \times 10^3$ rPGCLCs isolated by FACS for ITGA6/EPCAM at D4 of aggregate differentiation were xenotransplanted into each recipient testicle together with a single cell suspension of irradiated mouse testicular carrier cells ($3.0 \times 10^4$). For somatic cells, the ITGA6/EPCAM double negative cells at D4 were transplanted into irradiated male mice as above with a $1.3 \times 10^3$–$1.1 \times 10^4$ cell range. Mouse transplantation studies using irradiated nude mice were approved by The University of Texas M.D. Anderson Cancer Center Animal Care and Use Committee. Mouse transplantation studies performed on busulfan-treated nude mice were approved by the IACUC committees of Magee-Womens Research Institute and the University of Pittsburgh School of Medicine.

**Homologous transplantation into monkey seminiferous tubules**. Day 8 rPGCLCs aggregates (0.1 mg/mL Collagenase, Type IV (Gibco) and then 0.05% trypsin (Gibco)) and riPSCs (0.1 mg/mL Collagenase, Type IV (Gibco)) were dissociated prior to transplantation. Both rPGCLCs and riPSCs were suspended in MEMα containing 10% FBS, trypan blue (0.4 mg/mL; Sigma-Aldrich), 20% (v/v) Optison ultrasound contrast agent (GE Healthcare, Waukesha, WI, USA), 1% antibiotic–antimycotic (a combination of penicillin, streptomycin, and amphotericin B; Gibco) and DNase I (0.1 mg/mL) in a total volume of 0.5 mL. Using ultrasound guidance, to locate the rete testis, riPSCs were injected into the right testis and rPGCLC to the left testis of Rhesus macaques depleted of germ cells by testicular irradiation with 7 Gy[16]. Either 9.8 million riPSCs or 4.2 million rPGCLCs were injected per testis. To possibly enhance the grafting of the transplanted cells, the monkeys were given hormone suppression treatment using a GnRH-antagonist Acyline (obtained from the Contraceptive Development Program of the NICHD, Rockville, MD, USA) for 2 months starting immediately after irradiation and until the time of transplantation[16]. To prevent T cell-mediated rejection of the grafted cells, transplant recipients were treated with human/mouse chimeric anti-CD154 IgG 5C8 (NIH Nonhuman Primate Reagent Resource, University of Massachusetts Medical School, Boston, MA) at 20 mg/kg on Days −1, 0, 3, 10, 18, 28, and monthly thereafter. Changes in testes sizes and serum testosterone levels were monitored during the post-transplantation period and the testes were harvested 7 months after transplantation. The sliced testes were fixed in 4% paraformaldehyde and after initial observation under fluorescence scope for any GFP signal, they were either embedded in OCT and cryosectioned for immunofluorescence analysis, or embedded in paraffin for histology. All rhesus macaque transplantation studies were approved by The University of Texas M.D. Anderson Cancer Center Animal Care and Use Committee.

## Data availability

All relevant data are included in the paper and/or its Supplementary Information files and are available from the authors. A Reporting Summary for this Article is available as a Supplementary Information file.

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

## Acknowledgements

This project was funded by support from P01HD075795 (K.E.O., A.T.C., G.S., M.L.M.). Rhesus CS12 embryo collections were supported by the grant P51 OD011092 (J.D.H. and A.T.C.). E.S. acknowledges the support of the Eli and Edythe Broad Center of Regenerative Medicine and Stem Cell Research at UCLA Training Program for Post-doctoral Fellows. We would also like to acknowledge the BSCRC FACS Core and the BSCRC Imagining Core. We would also like to acknowledge Thein T. Phan for help with the xenotransplantation experiments. E.J.R. was supported by the National Institute of General Medical Sciences of the National Institutes of Health under award number R25GM055052 awarded to T. Hasson. Human fetal tissue research is supported by a grant to Ian Glass at the University of Washington Birth Defects laboratory 5R24HD000836-53.

## Author contributions

E.S. conceived and performed experiments and wrote the manuscript. D.C. performed FACS on rPGCs and rPGCLCs and prepared cells for transplant. E.J.R. performed immunofluorescence experiments on rPGCs and rPGCLCs. M.S., Z.W., and T.N.L. performed xenotransplantation experiments. K.A.P. and J.M.M. helped with homologous transplantations in rhesus macaques. R.C.T. performed dose and radiation field planning on each individual monkey. J.D.H. conceived experiments, maintained rhesus macaque IACUC approval for time mated breedings, and oversight of rhesus macaque work in Oregon. M.L.M. performed experiments, maintained IACUC approval, and oversight of mouse xenotransplantation and rhesus macaque transplantation in Houston. K.E.O. conceived the experiments, performed homologous rhesus macaque transplants. G.S. conceived and performed experiments, maintained IACUC approval, and oversight of mouse and rhesus macaque work in Houston. A.T.C. conceived the experiments, performed experiments, maintained Institutional Biosafety Approval for rhesus macaque riPSCs/rPGCLCs work and wrote the manuscript.

## Additional information

**Competing interests:** The authors declare no competing interests.

