## [Peer Review File · Nature Communications]

Reviewers' comments:

Reviewer #1 (Remarks to the Author):

The paper by Sosa and colleagues present transplantation experiments carried out in mice and rhesus macaques to determine the *in vivo* differentiation capacity of rPGCL cells. These experiments, performed for the first time in these species using PGCL aggregates, show that they can progress to VASA and MAGEA4 positive cells after transplantation in both species, however they do not express ENO2.

The paper is well written (except for minor issues detailed below) and the referencing is adequate. The conclusions are fair and the information presented is novel, and although the transplantation into rhesus monkey is limited to a few individuals, no statistical analysis was given for this small experiment. Clearly this is a major limitation of the model, and to gain a better understanding of why complete spermatogenesis was not achieved would need a larger scale study. For instance, it was not discussed whether a longer period of cellular maturation after transplantation would be needed in primates compared to mice. Following from examples in other tissues, for example differentiating human brain from ESC cells takes almost 9 months, one has to wonder whether achieving spermatogenesis in 7 months is sufficient time, and whether this has to be considered when trying to model this process *in vivo*.

Specific comments:

Line 181-190: The data using double immunostaining are of limited value without quantification data. Since the data presented in figure 2 is based on triple IF (SOX17/PRDM1/TFAP2C), I would suggest to remove supplementary figure 3a/b. Instead it would be more important to have a clear definition of the kinetic of gene expression of this triad of genes during the first 2-3 days after PGC induction. In *Cynomolgus* Monkey it seems TFAP2C plays a different role to that described for human PGC specification. In this context information from the Rhesus monkey is very relevant and could contribute to address this controversy in the field. Thus I would important that the authors provide detailed information for TFAP2C expression (quantification of cells observed) during the first 2 days after induction, and whether it was found co-expressed with other markers or not. At the moment the information is not possible to assess based on the pictures provided.

Quantification of this information can help clarify how efficient the induction is and the proportions of cells showing specific gene expression combinations.

Line 200-219: Epigenetic reprogramming of rPGCL cells seems, based on 5mC staining, that it has not started even after 8 days after induction. This is stark contrast to previous reports in hPGCL and cynomolgus PGCL cells, which show reduction in 5mC by day 4, but more importantly, 5hmC is strongly upregulated at this stage. Because of the close link between DNA demethylation and 5hmC, assessment of this mark is more meaningful for characterizing epigenetic reprogramming. To better assess the type of cell that is produced under the culture conditions used, it would be important to have a more comprehensive characterization of the epigenetic profile by including 5hmC staining after 4 and 8 days in the rhesus PGCL cells.

Figures:

Figure 1 legend:

1a: description of the legend does not reflect what the figure shows. A description of what the square shows should be indicated in the legend.

1c: Abbreviations in 1c are not included in the legend, but are in the results section. Figures should be standalone items and the legend should provide all the information shown. Please add the description of the abbreviations to the legend rather than in the results.

Figure 2: This figure shows dashed lines that are not described in the legend nor in the text. What are these trying to show?

Reviewer #2 (Remarks to the Author):

The manuscript by Sosa et al. demonstrates derivation of primordial germ cell-like cells (rPGCLCs) from rhesus macaque induced pluripotent stem cells (riPSCs), and differentiation of riPGCLCs by either transplantation into mouse testes (xenotransplantation) or rhesus macaque testes (homologous transplantation). The authors first determined that riPGCLCs in culture were equivalent to early rPGCs that do not expressed VASA, a marker gene of late PGCs. Upon xenotransplantation or homologous transplantation, riPGCLCs settled on the basement membrane and started to express VASA and MAGEA4, but not ENO2. It is also shown that riPGCLCs did not form teratoma, in contrast to frequent formation of teratoma in transplantation of riPSCs into the testes. These findings demonstrate that riPGCLCs have a potential to differentiate into a later stage of PGC development, and that xenotransplantation may be a good tool for evaluation, to some extent, of functionality of in vitro derived germ cells.

It seems that this manuscript provides important findings of differentiation capacity of riPGCLCs. These findings will help a deeper understanding of not only germ cell development in nonhuman primates but also in vitro gametogenesis from a technological point of view. There is, however, a slight concern: what are the cells actually in the testes? Taking account into the following suggestions, the author should add appropriate data to the manuscript.

Major concerns,

1. riPGCLCs in the testes expressed VASA and MAGEA4, which, as authors revealed, are markers of late PGCs in rhesus embryos. But they did not express ENO2, which is a more reliable marker for rhesus spermatogonia, meaning that riPGCLCs arrest at a later PGC stage. What is the stage they arrested at? There are several criteria to determine the stage, such as 5mC and cell cycle (to see prospermatogonia).
2. Is ENO2 expression really undetectable in the homologous transplant? Data in Figure 4i is ambiguous, as faint signals (background?) came up. Regarding to this concern, the author should show the number of GFP-positive cells counted for analysis of ENO2 expression.

Minor concerns

1. L152 (white dot circles) ?, instead of (white arrow head)
2. It would become more readable, if the authors add a sentence explaining "NHP".
3. The authors could have purified PGCLCs before transplantation. It would be better to mention the reason the author did not.
4. Scale bars in Fig 1b, Fig 2a and Fig5d.

Reviewer #3 (Remarks to the Author):

In this manuscript, Sosa and coworkers report an approach to induce primordial germ cell fate from rhesus macaque induced pluripotent stem cells in vitro. Furthermore, they show that after transplanted into adult mouse and monkey seminiferous tubules, the induced PGCLCs can overcome the major bottleneck on PGCLC maturation, and differentiate into the late PGCs and spermatogonia-like cells stage. This work provides the possibility to identify a chemically defined conditions to support the differentiation of immature PGCLC to VASA positive late stage PGCLC. The findings are overall interesting. However, several concerns should be addressed before consideration for publication.

1. The percentage of EpCAM and ITGA double positive cells during in vitro induction from Day1-8 should be shown, which helps to understand the effect of induction time on PGC fate decision.
2. In this report, the author transplanted Day 8 unsorted aggregate cells but not defined PGCLCs

into adult mouse and nonhuman primate seminiferous tubules and observed VASA/MAGEA4 positive cells emerged, however, that does not necessarily mean the VASA/MAGEA4 positive cells were derived from PGCLCs.

3. In previous studies, transplantation of primate pluripotent stem cells into mouse seminiferous tubules result in the differentiation of VASA positive cell, it suggested that the adult gonadal niche have the potential to induce germ cell fate from a pluripotent state. However, in this report, when monkey iPSCs were transplanted in to mouse and monkey testicles, no VASA positive cell could be detected, it is interesting to investigate or discuss the causes for those differences.

4. In this report, the author claimed adult gonadal niche helped to induce the immature PGCLCs commit to differentiate towards late PGCs and spermatogonia-like cells. However, there is no data showing weather the epigenetic reprogramming existes in these transition as well as the global demethylation from immature PGCs to late stage PGCs.

5. Seven months after transplantation, monkey testis were harvested and detected for the GFP signal. It is necessary to identify weather spermatogenesis occurs in these monkeys during these 7 months.

Point-by-point reply to the reviewers' comments:

We thank the reviewers for their thorough evaluation of the manuscript and recommendations to improve the quality and impact. In response to the reviewer's comments, we believe the manuscript has been improved. Changes to the manuscript are highlighted in yellow in the text. We have also added new Figures, Figure 5 and Supplementary Fig. 5 to address the reviewer's comments.

REVIEWER COMMENTS:

Reviewer #1 (Remarks to the Author):

The paper by Sosa and colleagues present transplantation experiments carried out in mice and rhesus macaques to determine the in vivo differentiation capacity of rPGCLC cells. These experiments, performed for the first time in these species using PGCL aggregates, show that they can progress to VASA and MAGEA4 positive cells after transplantation in both species, however they do not express ENO2.

The paper is well written (except for minor issues detailed below) and the referencing is adequate. The conclusions are fair and the information presented is novel, and although the transplantation into rhesus monkey is limited to a few individuals, no statistical analysis was given for this small experiment. Clearly this is a major limitation of the model, and to gain a better understanding of why complete spermatogenesis was not achieved would need a larger scale study. For instance, it was not discussed whether a longer period of cellular maturation after transplantation would be needed in primates compared to mice. Following from examples in other tissues, for example differentiating human brain from ESC cells takes almost 9 months, one has to wonder whether achieving spermatogenesis in 7 months is sufficient time, and whether this has to be considered when trying to model this process in vivo.

Response: We thank the reviewer for their time and appreciate the very positive comments!

1. Comment:

Line 181-190: The data using double immunostaining are of limited value without quantification data. Since the data presented in figure 2 is based on triple IF (SOX17/PRDM1/TFAP2C), I would suggest to remove supplementary figure 3a/b. Instead it would be more important to have a clear definition of the kinetic of gene expression of this triad of genes during the first 2-3 days after PGC induction. In Cynomolgus Monkey it seems TFAP2C plays a different role to that described for human PGC specification. In this context information from the Rhesus monkey is very relevant and could contribute to address this controversy in the field. Thus I would important that the authors provide detailed information for TFAP2C expression (quantification of cells observed) during the first 2 days after induction, and whether it was found co-expressed with other markers or not. At the moment the information is not possible to assess based on the pictures provided. Quantification of this information can help clarify how efficient the induction is and the proportions of cells showing specific gene expression combinations.

Response: We appreciate the suggestions to (1) quantify the kinetics of TFAP2C expression in Day 1, 2, 3, and 4 aggregates and (2) provide detailed information on the co-expression of TFAP2C with PRDM1 and/or SOX17. We have performed new experiments and updated Figure 2 to include new graphs (Fig. 2c and Fig.2d) as well as updated our results section and the related discussion. Lastly, we have also updated the methods section to include a new section called "Image Analysis"

2. Comment:

Line 200-219: Epigenetic reprogramming of rPGCL cells seems, based on 5mC staining, that it has not started even after 8 days after induction. This is stark contrast to previous reports in hPGCL and cynomolgus PGCL cells, which show reduction in 5mC by day 4, but more importantly, 5hmC is strongly upregulated at this stage. Because of the close link between DNA demethylation and 5hmC, assessment of this mark is more meaningful for characterizing epigenetic reprogramming. To better assess the type of cell that is produced under the culture conditions used, it would be important to have a more comprehensive characterization of the epigenetic profile by including 5hmC staining after 4 and 8 days in the rhesus PGCL cells.

Response: The reviewer brings up an important point and to address this we performed new experiments to characterize the epigenetic profiles (5hmC and 5mC) in rPGCLCs at Day 2, 4, and 8 of aggregate differentiation. We also characterized rPGCs at D28 and D50 for 5hmC and 5mC as in vivo comparisons. These new results can be found in Figure 3 (Fig. 3d and Fig.3e).

3. Comment:

Figure 1 legend:

1a: description of the legend does not reflect what the figure shows. A description of what the square shows should be indicated in the legend.

Response: We have added text to the Figure 1 and clarify that the dotted line highlights the region where we found in vivo rPGCs, as they migrated from the hindgut towards the genital ridge epithelium.

4. Comment:

Figure 1 legend:

1C: Abbreviations in 1c are not included in the legend, but are in the results section. Figures should be standalone items and the legend should provide all the information shown. Please add the description of the abbreviations to the legend rather than in the results.

Response: We have added descriptions of the abbreviations to the legends.

5. Comment:

Figure 2: This figure shows dashed lines that are not described in the legend nor in the text. What are these trying to show?

Response: We included text to the legend, which clarifies that the dotted lines highlight clusters of putative rPGCLCs.

Reviewer #2 (Remarks to the Author):

The manuscript by Sosa et al. demonstrates derivation of primordial germ cell-like cells (rPGCLCs) from rhesus macaque induced pluripotent stem cells (riPSCs), and differentiation of riPGCLCs by either transplantation into mouse testes (xenotransplantation) or rhesus macaque testes (homologous transplantation). The authors first determined that riPGCLCs in culture were equivalent to early riPGCs that do not expressed VASA, a marker gene of late PGCs. Upon xenotransplantation or homologous transplantation, riPGCLCs settled on the basement membrane and started to express VASA and MAGEA4, but not ENO2. It is also shown that riPGCLCs did not form teratoma, in contrast to frequent formation of teratoma in transplantation of riPSCs into the testes. These findings demonstrate that riPGCLCs have a potential to differentiate into a later stage of PGC development, and that xenotransplantation may be a good tool for evaluation, to some extent, of functionality of in vitro derived germ cells.

It seems that this manuscript provides important findings of differentiation capacity of riPGCLCs. These findings will help a deeper understanding of not only germ cell development in nonhuman primates but also in vitro gametogenesis from a technological point of view. There is, however, a slight concern: what are the cells actually in the testes? Taking account into the following suggestions, the author should add appropriate data to the manuscript.

Response: We thank the reviewer for their time and appreciate the very positive comments!

1. Comment:

riPGCLCs in the testes expressed VASA and MAGEA4, which, as authors revealed, are markers of late PGCs in rhesus embryos. But they did not express ENO2, which is a more reliable marker for rhesus spermatogonia, meaning that riPGCLCs arrest at a later PGC stage. What is the stage they arrested at? There are several criteria to determine the stage, such as 5mC and cell cycle (to see prospermatogonia).

Response: We appreciate the comment and performed new experiments to test this. These results can be found in new Figures including, Fig 5 (previous Figure 5 is now Figure 6) and the corresponding Supplementary Figure 5. This new figure also includes a detailed characterization of Ki67 expression (as an indicator of the cell cycle) in rPGCLCs at D4 and D8 of aggregate differentiation, rPGCs at D28 and D50 (Fig 5b, 5c) and in the transplanted cells (Fig. 5d), Also, we have updated the methods section to show how fraction of Ki67+ cells were quantified in the germ cells and somatic cells.

2. Comment:

Is ENO2 expression really undetectable in the homologous transplant? Data in Figure 4i is ambiguous, as faint signals (background?) came up. Regarding to this concern, the author should show the number of GFP-positive cells counted for analysis of ENO2 expression.

Response: To address this comment we changed the text of the results section to reflect the number of Nhp⁺ cells that were counted and assessed for the co-expression of ENO2.

3. Comment:

L152 (white dot circles)?, instead of (white arrow head)

Response: We changed the text to Figure 4's legend, which clarifies that the white dotted lines highlight the basement membrane of the tubules, which may (white arrows) or may not (no arrows) contain non-human primate positive germ cells.

4. Comment:

It would become more readable, if the authors add a sentence explaining "NHP".

Response: We changed the text of the results section, which is the first instance of the use of the term non-human primate, which is then abbreviated to Nhp, and used as Nhp in the text from then on.

5. Comment:

The authors could have purified PGCLCs before transplantation. It would be better to mention the reason the author did not.

To directly address your comment we used FACS to isolate GFP+ labeled rPGCLCs (using EPCAM/ITGA6), and GFP+ labeled somatic cells (negative for EPCAM and ITGA6), and transplanted the sorted populations into different testicles. Our results show that GFP signal is detected in testicles transplanted with rPGCLCs isolated by FACS prior to xenotransplantation, whereas no GFP+ signal is detectable in testicles transplanted with somatic cells. This result can be found in Supplementary Fig. 4h.

6. Comment:

Scale bars in Fig 1b, Fig 2a and Fig5d.

Response: We have added scale bars to Figure 1b, Figure 2a, and Figure 6d (formerly Figure 5d) and added the relevant text to the legend.

Reviewer #3 (Remarks to the Author):

In this manuscript, Sosa and coworkers report an approach to induce primordial germ cell fate from rhesus macaque induced pluripotent stem cells in vitro. Furthermore, they show that after transplanted into adult mouse and monkey seminiferous tubules, the induced PGCLCs can overcome the major bottleneck on PGCLC maturation, and differentiate into the late PGCs and spermatogonia-like cells stage. This work provides the possibility to identify a chemically defined conditions to support the differentiation of immature PGCLC to VASA positive late stage PGCLC. The findings are overall interesting. However,

several concerns should be addressed before consideration for publication.

Response: We thank the reviewer for their time and appreciate the very positive comments!

1. Comment:

The percentage of EpCAM and ITGA double positive cells during in vitro induction from Day1-8 should be shown, which helps to understand the effect of induction time on PGC fate decision.

Response: We appreciate the reviewer's interest in the number of EPCAM and ITGA6 double positive cells that are induced from day 1, 2, 4, and 8 aggregates. We performed new experiments and have included the data in Figure 3a and have updated the text of the results.

2. Comment:

In this report, the author transplanted Day 8 unsorted aggregate cells but not defined PGCLCs into adult mouse and nonhuman primate seminiferous tubules and observed VASA/MAGEA4 positive cells emerged, however, that does not necessarily mean the VASA/MAGEA4 positive cells were derived from PGCLCs.

Response: We agree with the reviewer that our original study design did not preclude that germ cells arose from non-rPGCLCs (EPCAM/ITGA6⁺) transplanted cells. Therefore, we designed experiments to test whether sorted PGCLCs (versus non-rPGCLCs) resulted in engraftment in mice and have included this data as an in Supplementary Fig. 4 (Supplementary Fig. 4h). Although we would have liked to also perform this experiment in the non human primate, the rPGC cell number at the corresponding stage ~D28) makes it unfeasible.

3. Comment:

In previous studies, transplantation of primate pluripotent stem cells into mouse seminiferous tubules result in the differentiation of VASA positive cell, it suggested that the adult gonadal niche have the potential to induce germ cell fate from a pluripotent state. However, in this report, when monkey iPSCs were transplanted in to mouse and monkey testicles, no VASA positive cell could be detected, it is interesting to investigate or discuss the causes for those differences.

Response: We appreciate the reviewer asking for clarification on why we do not get the same results as previous reports. We believe that these differences may reflect the differences between human and non-human primate iPSCs and have stated this in the discussion.

4. Comment:

In this report, the author claimed adult gonadal niche helped to induce the immature PGCLCs commit to differentiate towards late PGCs and spermatogonia-like cells. However, there is no data showing weather the epigenetic reprogramming exists in these transition as well as the global demethylation from immature PGCs to late stage PGCs.

Response: Thank you for your comment, which is similar to reviewer 2's. To address this comment, we performed new experiments by characterizing the epigenetic profiles (5hmC and 5mC) in rPGCLCs at Day 2, 4, and 8 of aggregate differentiation. We also characterized rPGCs at D28 and D50 for 5hmC and 5mC. We have updated our results section to include this new data in Figure 3 (Fig. 3d and Fig.3e). Lastly, we examined the epigenetic profiles of xenotransplanted cells and this can be found in the new Figure 5a and Supplementary Fig. 5a-b.

5. Comment:

Seven months after transplantation, monkey testis were harvested and detected for the GFP signal. It is necessary to identify weather spermatogenesis occurs in these monkeys during these 7 months.

Response: Thank you for your comment, which is similar to reviewer 2's concern. We agree that these experiments will be important and are necessary and will be the focus of future studies.

REVIEWERS' COMMENTS:

Reviewer #1 (Remarks to the Author):

The revised version of the ms. by Sosa et al., provides additional information that helps with understanding the efficiency of rPGCLC differentiation in vitro, as well as the molecular features of these cells. The new information on 5hmC content in these cells is also very informative, as it demonstrates distinct properties between pre and post migratory PGCs.

The revised version also addresses all the other minor concerns.

The new figures provided are well presented and the revised structure of the manuscript as well as the results/discussion are well written.

I think this paper is a valuable contribution to the scientific community and recommend its publication in its current form.

Ramiro Alberio

Reviewer #2 (Remarks to the Author):

The authors revised the manuscript with new results from transplantation analyses. With immunofluorescence analyses of 5mC, 5hmC and cell cycle, the authors conclude that PGCLCs differentiated into a stage corresponding to later PGCs that express VASA and MAGEA4. The authors also concluded that epigenetic reprogramming was incomplete, since PGCLCs in the transplant were 5hmC-positive but Ki67-negative.

Although the stage, at which PGCLCs arrested in the transplant, is still not entirely clear, these findings will provide useful information of a differentiation process of in vitro-derived germ cells. A recent report showed that differentiation of human PGCLCs took long time (~12weeks) to become oogonia (Yamashiro et al 2018 Science). Therefore, it may take a long time to differentiate PGCLCs to spermatogonia in Rhesus Macaque. Nevertheless, the authors addressed, at least, to all claims I suggested and the results are evaluated adequately. Therefore, it is feasible to publish this manuscript in Nature Communications.

Reviewer #3 (Remarks to the Author):

The authors have satisfactorily addressed nearly all of my concerns, and I have no further questions.

Point-by-point reply to Reviewer Request

We thank the Reviewers for all of their detailed comments and recommendations that were given to us for the manuscript entitled "Differentiation of primate primordial germ cell-like cells following transplantation into the adult gonadal niche." We believe that the reviewers identified important areas that required improvement. After completion of the suggested edits we believe that the quality and impact of the manuscript has significantly improved. Below please find the Reviewer comments and our point by point responses.

REVIEWERS' COMMENTS:

Reviewer #1 (Remarks to the Author):

The revised version of the ms. by Sosa et al., provides additional information that helps with understanding the efficiency of rPGCLC differentiation in vitro, as well as the molecular features of these cells. The new information on 5hmC content in these cells is also very informative, as it demonstrates distinct properties between pre and post migratory PGCs.

The revised version also addresses all the other minor concerns.

The new figures provided are well presented and the revised structure of the manuscript as well as the results/discussion are well written.

I think this paper is a valuable contribution to the scientific community and recommend its publication in its current form.

Ramiro Alberio

Response: We thank the reviewer for all of their suggestions as well as their thorough review of our manuscript.

Reviewer #2 (Remarks to the Author):

The authors revised the manuscript with new results from transplantation analyses. With immunofluorescence analyses of 5mC, 5hmC and cell cycle, the authors conclude that PGCLCs differentiated into a stage corresponding to later PGCs that express VASA and MAGEA4. The authors also concluded that epigenetic reprogramming was incomplete, since PGCLCs in the transplant were 5hmC-positive but Ki67-negative.

Although the stage, at which PGCLCs arrested in the transplant, is still not entirely clear, these findings will provide useful information of a differentiation process of in vitro-derived germ cells. A recent report showed that differentiation of human PGCLCs took long time (~12weeks) to become oogonia (Yamashiro et al 2018 Science). Therefore, it may take a long time to differentiate PGCLCs to spermatogonia in Rhesus Macaque. Nevertheless, the authors addressed, at least, to all claims I suggested and the results are evaluated adequately. Therefore, it is feasible to publish this manuscript in Nature Communications.

Response: We thank the reviewer for their comments and suggestions and appreciate their critical review of our manuscript

Reviewer #3 (Remarks to the Author):

The authors have satisfactorily addressed nearly all of my concerns, and I have no further questions.

Response: We are pleased we have satisfactorily addressed the reviewers concerns and we are grateful for their detailed review of our manuscript.